# An empirical analysis of the demand for family planning satisfied by modern methods among married or in-union women in Nigeria: Application of multilevel binomial logistic modelling technique

Emomine Odjesa[1]*, Friday Ebhodaghe Okonofua[1,2,3]

**1** Centre of Excellence in Reproductive Health Innovation (CERHI), University of Benin, Benin City, Edo State, Nigeria, **2** Department of Obstetrics and Gynaecology, University of Benin, University of Benin Teaching Hospital, Benin City, Edo State, Nigeria, **3** Women's Health and Action Research Centre (WHARC), Benin City, Edo State, Nigeria

* odjesaemomine@yahoo.com

**Data Availability Statement:** The data file is publicly available from the DHS Program website

## Abstract

### Background

Given the health and economic benefits of family planning (FP), Nigeria's very low demand for FP satisfied by modern methods (mDFPS) of less than 50% is therefore a major public health concern, especially considering the global target aimed at achieving an mDFPS of at least 75% by year 2030 for all countries. In view of this, together with recognising the possible contextual nature of health outcomes, this study aimed to empirically analyse the mDFPS among married or in-union women of reproductive age (WRA) in Nigeria.

### Materials and methods

A multilevel binomial logistic model with two levels of analysis was used: individual and community levels. Secondary cross-sectional data were obtained from the 2018 Nigeria Demographic and Health Survey, and analyses were performed using Stata 15.0. The analytical sample size was 9,122 WRA nested in a total of 1,072 communities.

### Results

The mDFPS was approximately 31.0%. The median odds ratio (MOR) estimated from the final multilevel model was 2.245, which was greater than the adjusted odds ratio (aOR) for most of the individual-level variables, suggesting that the unexplained/residual between-community variation in terms of the odds of women having their mDFPS was more relevant than the regression effect of most of the individual-level variables. This was with the exception of the regression effects of the following individual-level variables: women's husbands that had higher education level in comparison to their counterparts who had husbands with no formal education (aOR = 2.539; 95% CI = 1.896 to 3.399; p<0.001); and women from the Yoruba ethnic group in comparison to their counterparts from the Hausa/Fulani/Kanuri

(https://dhsprogram.com/data/dataset/Nigeria_Standard-DHS_2018.cfm?flag=0) upon receiving a written approval to a formal written request made by the researcher. Institutional access is thus needed, after which the data can be freely accessed.

**Funding:** The authors received no specific funding for this work.

**Competing interests:** The authors have declared that no competing interests exist.

ethnic group (aOR = 2.484; 95% CI = 1.654 to 3.731; p value<0.001). However, other individual-level variables with positive statistically significant regression effects on mDFPS were: women who mentioned that money for accessing health care was not a problem; women's empowerment in relation to the visitation of family and relatives; and women being exposed to FP messages through various media sources, all in comparison to their respective counterparts. On the other hand, at the community level, women in communities where a high percentage of them had at least a secondary education had statistically significant greater odds of having mDFPS than women in communities with lower education levels (aOR = 1.584; 95% CI = 1.259 to 1.991; p<0.001). We found similar findings regarding women residing in communities with exposure to FP messages through various media sources. However, using the 80% interval ORs (80% IORs) as a supplemental statistical measure for further understanding the regression effects of community-level variables showed that all of the 80% IORs had a value of '1', signifying considerable uncertainty in the regression effects of all community-level variables due to the substantial residual variation existing between communities.

## Conclusions

Our study showed that to achieve the dire increase in mDFPS in Nigeria, policy interventions aimed at improving the education level of both females and males, especially beyond the secondary school level, should be implemented. Additionally, all of the various media sources should be extensively utilised, both at the individual and the community level, by the Nigerian government to spread information on the importance of women having their mDFPS.

## Introduction

The United Nations (UN) Sustainable Development Goals (SDG) have the 1st indicator of the 7th target of the 3rd goal (that is, the SDG indicator 3.7.1) as the demand for family planning satisfied by modern methods (mDFPS) [1]. This is the only indicator used to represent family planning (FP) from the SDG list of indicators and hence to track its progress in relation to the achievement of sustainable development [1], with a global benchmark or target to achieve at least 75% coverage in the mDFPS by the year 2030 in various countries [2]. The indicator 'mDFPS' represents the percentage of women of reproductive age (WRA), that is, those 15 to 49 years old, who are satisfying or meeting their demand for family planning (DFP) with the use of modern contraceptives (MC) [3,4]. DFP, on the other hand, represents the total need for FP among the WRA—that is, the sum of the need for spacing a/next births and the need for limiting births, including those who do not want children at all, whether these needs are met/satisfied with contraceptives or not [3–5].

The indicator 'mDFPS' is thus said to be a better representation of modern contraceptive use (MCU) since it concentrates on WRA with a need for FP (NFP) [6]. This is in contrast to other contraceptive indicators which concentrate on all WRA, irrespective of their NFP [6–8]. For example, the modern contraceptive prevalence rate (mCPR) or the contraceptive prevalence rate(CPR), which were used by the UN Millennium Development Goals (MDGs) as indicators to assess FP progress [6–8]. The contraceptive indicator 'mDFPS' is thus said to better reflect the voluntariness associated with the UN description of reproductive rights and fertility

control, which should, however, be done responsibly [6,9,10]. According to the 2018 Nigeria Demographic and Health Survey (NDHS), the mDFPS among married or in-union and sexually active unmarried women of reproductive age was 33.9% and 32.5%, respectively [5]. These values are considered very low since they are less than 50% [11]. In addition, according to the data from previous NDHS reports, the mDFPS in Nigeria has been consistently very low, although it has gradually increased over the years, with a value of 13.1% among married or in-union WRA, as obtained from the 1990 NDHS [5,12–16].

It is important to note that since contraceptives, particularly modern methods, are a main proximate determinant of fertility [17,18], Nigeria's consistently very low mDFPS has thus contributed to the country's persistently high total fertility rate (TFR), with a value of 5.3 live births for each woman in 2019 [19]. In fact, the TFR in Nigeria has remained high at approximately 6.0 live births for each woman from 1990 to 2016, implying no significant reduction in the TFR across the years [19]. Note that a high TFR is defined as more than 5 live births for each woman [20]. Furthermore, an increase in mDFPS, through its reducing effect on the TFR, will lead to a reduction in total population count (TPC), particularly a reduction in the child-dependent population as a result of the reduction in birth rates [21,22]. This will help Nigeria achieve demographic dividends, which can be utilised for economic growth and development through the implementation of appropriate government policies, with an associated improvement in the economic status of individuals, households, and the country [23,24]. This is particularly important since, with a population of approximately 224 million people in 2023, Nigeria is presently the 6th most populated country worldwide and the most populated country in Africa [25].

Therefore, taken together, there is a dire need to increase the mDFPS in Nigeria. This will also increase Nigeria's possibility of meeting the SDG indicator 3.7.1 target/benchmark by the year 2030. This can be achieved through a proper understanding of the possible factors associated with an increase in the mDFPS in Nigeria. However, health outcomes, including contraceptive use, are often said to be contextual in nature [26,27]. This means that the clusters in which people reside, which are usually social, environmental, or administrative spaces such as communities, have unique characteristics that possibly influence health outcomes, including MCU [26,27]. These characteristics, both measured and unmeasured, include gender relations; social norms or traditions; cultural and religious laws (CRLs) such as Sharia law and the non-domestication of the Child Rights Act (CRA) within states; socioeconomic characteristics; and the availability/nonavailability of infrastructures [26–28].

All of these characteristics will therefore vary from one community to another [26]. People who reside within a community are thus possibly exposed to/affected by similar characteristics as opposed to those who reside in a different community with different characteristics [26]. Therefore, those who reside within the same community may behave in similar ways with possibly similar risk/likelihood of a particular health outcome occurring [26]. This will result in a between-community variance and a within-community variance in the odds/likelihood/risk of the health outcome being achieved [26].

Investigating the contextual nature of the mDFPS in Nigeria will help provide empirical evidence for not only the formulation and application of government policies centred on individual women but also provide policies that are cluster-centred, either residential, administrative, or geographical clusters. This approach will thus aid in reducing health inequalities not only between individual women within a cluster but also between clusters, thereby leading to an improvement in this health outcome nationally. This is particularly important in Nigeria, as she is a culturally, ethnically, religiously, economically, and legally pluralistic society [5,28].

Few empirical studies, both inside and outside of Nigeria, have used the indicator 'mDFPS' as the outcome variable of interest in their different studies [3,4,7,29–31]. Only the studies by Fagbamigbe et al. [3] and Ewerling et al. [4] were carried out within the Nigerian setting or

included Nigerian women. However, for both of these studies, only descriptive statistical analyses were carried out [3,4]. Thus, it could not be determined from the results of these studies if the associations found between the different independent variables and the mDFPS occurred by chance or if these were true effects [3,4]. It can thus be concluded that within Nigerian settings, further inferential statistical analysis is needed. Therefore, against this background, this quantitative study was carried out to determine the individual and community-level factors associated with Nigerian married or in-union WRA satisfying their DFP with MC. Additionally, to determine the between-community variance and the within-community variance, in the odd/likelihood of the Nigerian married or in-union WRA satisfying their DFP with MC, and how these associated factors compare with this between-community variance.

## Materials and methods

### Conceptual framework

This study's conceptual framework was adapted from the Health Belief Model (HBM) [32,33]. The HBM explains people's behavioural patterns in relation to their adoption of preventive strategies for diseases or unfavourable health outcomes [32,33]. This, therefore, also includes the use of MC to prevent unintended pregnancies by women with an identified NFP [32,33]. According to the HBM, women's use of MC is based on two factors—women's perception of the threat of an unintended pregnancy occurring (together with its sequelae) and the perception of the costs (including the barriers) and benefits of using MC [32,33]. These two factors can be modified by various enabling factors (such as socioeconomic factors and demographic factors) and internal and external cues of action (such as receiving information about the use of FP from various media sources and health workers) operating at both the individual and contextual levels [32,33]. Further details about the HBM can be found elsewhere [32,33].

### Study setting

Nigeria is a sovereign nation situated on the western side of the African continent, with her large population distributed across 774 Local Government Areas(LGA), which are grouped into 36 states and the Federal Capital Territory (FCT), Abuja [26]. The states are also in turn grouped into six geopolitical zones—North Central (NC), North East (NE), North West (NW), South West (SW), South East (SE), and South South (SS) [5,26]. According to the World Bank Group, in 2022, approximately 54% of the Nigerian population lived in urban areas [19]. Additionally, although Nigeria is considered to have the largest economy in Africa [26,34], with a Gross Domestic Product (GDP) per capita of $4,963 in 2022 based on purchasing power parity (PPP) in constant prices of 2017, approximately 30.9% of the population lived below the international poverty line of $2.15 per day in 2018, still based on the 2017 PPP [19]. However, differences occur across the Northern and Southern states, and hence the Northern and Southern geopolitical zones, and across the urban and rural areas in terms of socioeconomic development, including poverty levels, with the Southern zones and the urban areas usually being more affluent than their respective counterparts [5,26].

In addition, the Nigerian population is said to be diverse in terms of religion and ethnicity [5,26,34]. For example, in 2018, 53.5% of the Nigerian population were Muslims, 45.9% were Christians, and the remaining 0.6% were either African traditional religionists, or affiliated with other religions apart from the aforementioned, or not affiliated with any religion at all [34]. The Northern geopolitical zones of Nigeria are dominated by Muslims, with Christians present as a sizeable minority in some parts of the Northern states [35]. On the other hand, the Southern geopolitical zones of Nigeria are dominated by Christians, with a sizeable number of Muslims in the Southern states [35]. A more even distribution of Muslims and Christians in

Nigeria occurs more in the NC geopolitical zone, popularly referred to as the middle belt, and in some parts of the SW zone [35].

Nigeria is also home to individuals from more than 250 ethnic groups within which there are distinct subgroups and communities [35,36]. This ethnic heterogeneity occurs across the country [36]. However, there are three main ethnic groups, and they are located mainly in distinct parts of the country. These ethnic groups, in addition to being the most populous, are also the most politically influential [37]. These are Yoruba, the dominant ethnic group in the SW geopolitical zone of Nigeria; Igbo, the dominant ethnic group in the SE zone of Nigeria; and Hausa and Fulani, which are the dominant ethnic groups in the Northern zone of Nigeria [37]. The latter constitute the largest ethnic group in Nigeria, with 36% of Nigerians identified as Hausa and Fulani in 2018 [34]. This is followed by the Yoruba ethnic group, with 15.5% of Nigerians identified therein and then 15.2% of Nigerians identified as Ibos [34].

It is also important to mention the Kanuri ethnic group, which constitutes the second largest ethnic group in the Northern geopolitical zone of Nigeria and is mainly found in the NE zone, with approximately 2.4% of Nigerians identified within this ethnic group in 2018 [34,35]. The other ethnic groups are called 'ethnic minorities' and are located in various parts of the Northern and Southern geopolitical zones of Nigeria [37]. Furthermore, this diversity in Nigeria's ethnic groups also leads to diversity in the indigenous spoken languages [34]. In fact, Nigeria is said to have more than 500 local languages, but Hausa, Igbo, and Yoruba are the major indigenous languages [34]. This implies that having three main ethnic groups translates into having three main local languages for Nigeria [34]. However, the country's official language is English [34].

Nigeria is also pluralistic legally, thereby causing contradictions regarding the same legal matters within the same country [28,36]. In relation to sexual and reproductive health (SRH), this is particularly true for Islamic or Sharia law [28]. The Islamic or Sharia law has been known to undermine improvements in SRH through, for example, having a different policy regarding the legal age of marriage and the legal age of consent for sex [28]. Note that the CRA, which was passed into law in 2003, establishes the minimum age for marriage in Nigeria for both sexes as 18 years, while the 2013 Nigeria's Sexual Offences Act establishes the age of sexual consent for both sexes as 18 years [28]. However, these acts will only apply to Nigerian states that have adopted them; that is, the incorporation of these acts into the individual state's legislation to give it force, which are usually the Southern states [28]. Due to opposition by the Supreme Council for Sharia, neither of these acts has been adopted by many Northern states, especially those in the NE and NW geopolitical zones, resulting in variation in the age of marriage and in the age of sexual consent for females within Nigeria [28]. The age of marriage and age of sexual context for females could thus be less than 18 years, or even defined as the age of puberty, in these Northern states that have not adopted these acts [28].

Therefore, taken together, wide disparities do occur between the Northern and Southern states in Nigeria and hence across the different geopolitical zones in Nigeria in terms of socio-economic development, cultural practices, and the utilisation of healthcare services [26]. These differences will thus also be noted across the different major ethnicities and the different major religions in Nigeria because these ethnic groups and religions are also divided across the Northern and Southern states and hence the geopolitical zones in Nigeria [26]. These differences are also observed across Nigerian rural and urban areas [5].

## Source of data

This study utilised secondary cross-sectional data obtained from the individual recode file of the 2018 NDHS; a population-based survey carried out in Nigeria and which at the time of this study was the recent Demographic and Health Survey (DHS) with data available [5]. The

individual recode file contains anonymised information on the socioeconomic, demographic, and reproductive characteristics, including contraceptive use, of Nigerian women and their husbands/partners at the individual and/or contextual levels [5]. The 2018 NDHS data were collected through a multistage stratified sampling technique, with stratification achieved through the separation of the 36 states in Nigeria and the Federal Capital Territory (FCT) into rural and urban areas, thereby producing 74 sampling strata [5]. The first sampling stage involved the random selection of 1,400 enumeration areas (EAs), which are referred to as clusters for the 2018 NDHS, as defined on the basis of the EAs used by the 2006 Nigeria Population and Housing Census (NPHC) frame [5]. The EAs are thus the primary sampling units (PSUs) [5]. The PSUs were used in this study to proxy communities, as in other empirical works [26].

The second sampling stage involved the random selection of 30 households from each EA, giving a total of 42,000 households [5]. However, the survey was successfully performed in 1,389 EAs and 40,427 households, with data collected from 41,821 WRA [5]. Therefore, the 2018NDHS has a multilevel/hierarchical/nested data structure with lower levels nested within higher levels [5]. More details about the sampling design of the 2018 NDHS can be found in the survey report [5]. In addition, note that the 2018 NDHS data are freely accessible online from the DHS website at https://dhsprogram.com/data/dataset/Nigeria_Standard-DHS_2018.cfm?flag=0 upon receiving a written approval to a formal written request [38]. No further ethical clearance was needed for this study because the data were anonymised.

## The study's analytical sample

Based on the aims of this study, concentration was done on the married or in-union WRA in Nigeria with a DFP (that is, the totality of NFP), who were also usual residents of the communities in which they were surveyed. This clearly depicts the eligibility criteria for the women to be included in this study and thus represents this study's analytical sample. Note that only the usual residents of the communities surveyed by the 2018 NDHS were concentrated on in this study because it allowed for examining the possible contextual influences on the likelihood of Nigerian married or in-union WRA with a DFP using MC. Fig 1 shows several exclusions made from the general 2018 DHS dataset to arrive at this study's analytical sample.

Note that in this study, identifying the women with an NFP, both met and unmet needs, was based on the revised description by Bradley et al. [39], with all the married or in-union WRA described as being sexually active. This revised description of women with an NFP by Bradley et al. [39] was also used by the 2018NDHS [5]. The women with an NFP according to Bradley et al. [39] are as follows:

a. fecund women who were using (or their husbands/partners were using) any modern or non-modern method of contraceptives at the time of the survey for spacing or limiting births—constituting met need for spacing births or met need for limiting births, respectively;

b. all the pregnant women whose pregnancies were either unwanted or mistimed at the time of the survey—constituting unmet need for limiting births or unmet need for spacing births, respectively;

c. all the postpartum amenorrhoeic women who were not using (or whose husbands/partners were not using) any modern or non-modern contraceptives at the time of the survey and whose latest birth(s) were either unwanted or mistimed—constituting unmet need for limiting births or unmet need for spacing births, respectively;

d. all other fecund women who were not using (or whose husbands/partners were not using) any modern or non-modern contraceptives at the time of the survey but who either wanted

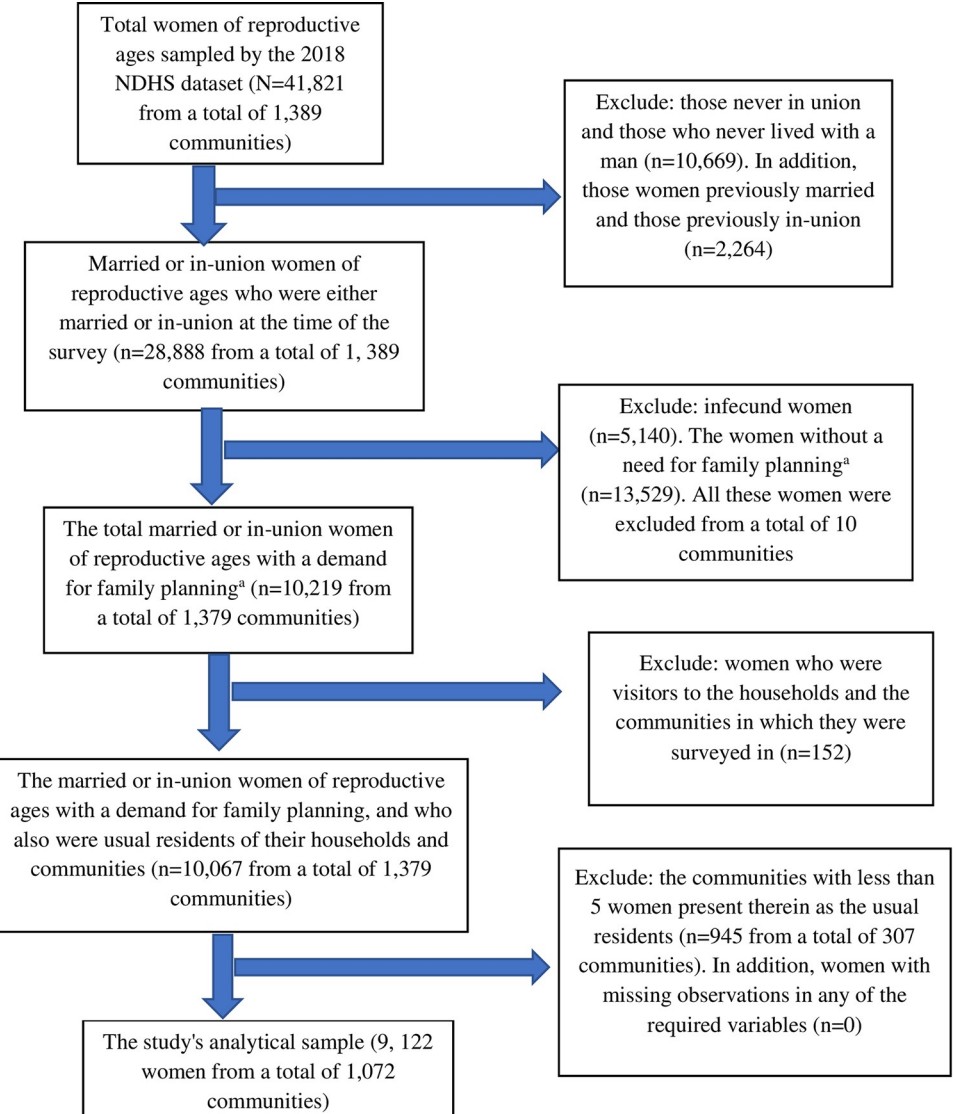

**Fig 1. Flow chart for obtaining this study's analytical sample.**

to stop childbearing or did not want children at all—constituting unmet need for limiting births; and

e. all the other fecund women who were not using (or whose husbands/partners were not using) any modern or non-modern contraceptives at the time of the survey but either wanted to postpone their next childbearing or delay their first child birth for at least two years—constituting unmet need for spacing births. Additionally, included in this category are the fecund women who want a child (that is, the first child) or another/next child, together with not using (or their husbands/partners not using) any modern or non-modern contraceptives at the time of the survey, but are undecided about the timing of having this child or those even undecided about wanting a child or another child.

Adding together all the women in categories 'a' to 'e' above gives the total NFP or DFP. Also, following Clark [40], all the communities with less than 5 WRA with a DFP as the usual

resident were eliminated so as to obtain valid and reliable regression estimates of all parameters, including random effects. The final unweighted analytical sample for this study was thus composed of 9,122 Nigerian married or in-union WRA with a DFP, all of which were usual residents of a total of 1,072 communities.

Additionally, note that since this study's analytical sample was obtained from the 2018 NDHS and because of the sample design of this NDHS, as explained previously, this study's analytical sample thus covers the entire territory of Nigeria. This analytical sample is therefore representative of the Nigerian population structure, together with its naturally occurring hierarchies. This thus allows for, using appropriate statistical analysis, the distinguishing of individual-level and community-level factors, together with the estimation of the different variances at the different levels, that can possibly influence the likelihood of married or in-union women having mDFPS. Also, it allows for, using appropriate statistical techniques, the generalisation of our result findings from this sample into the Nigerian population. Thus, this analytical sample can be used to meet the aims of this study.

## Measures

**Outcome or dependent variable.** Based on the aim of this study, the outcome or dependent variable represents the type of contraceptive (or lack of it thereof) used by the WRA with a DFP. It is a categorical (binary) variable with the following two subcategories:

(a) The DFP currently met with women's use (or their husbands'/partners' use) of MC, regardless of the modern type. This subcategory will be referred to as the 'mDFPS'; and (b) DFP currently met with women's use (or their partners'/husbands' use) of non-modern methods or not met with any method at all, whether it is modern or non-modern. This subcategory will be referred to as 'non-mDFPS'. The latter subcategory is the reference subcategory. Note that the word 'currently' is used here to represent the information collected at the time the 2018NDHS was carried out.

Furthermore, following Hubacher and Trussell [41], the MC used in this study were as follows: female sterilisation; male sterilisation; intrauterine device (IUD); implants; injectables; emergency contraception; pills; vaginal rings; male condoms; female condoms; diaphragms; caps; and spermicidal agents such as cream, foam/sponge, and jelly. Contraceptive methods outside of these were thus classified as non-modern methods [41]. These are listed as follows: lactational amenorrhoea method (LAM); withdrawal method; periodic abstinence; and all the fertility awareness-based methods, such as basal body temperature methods, symptom-thermal methods, standard days methods (SDM), and two-day methods [41]. According to the 2018 NDHS, there are also folkloric methods, such as herb, bead, amulet, and douching methods [5]. The users of these folkloric methods were grouped in this study among those not using any contraceptive method at all. This is because these methods do not have a scientifically known effectiveness level [42].

**Independent or explanatory variables.** The independent variables used in this study were all categorical and included the following variables: sociodemographic variables; the presence or absence of CRLs; health institution access variables (including monetary access); household economic variables; women's empowerment variables; and women's knowledge of FP as assessed by their exposure to FP messages through various media sources. These independent variables were obtained from both the empirical literature review and the conceptual framework for this study.

## Type of empirical model

Since the dependent variable is a binary variable, while still considering the aims of this study together with the use of the NDHS data with its multilevel structure, the model of choice was

the random intercept mixed effect multilevel multivariable binomial logistic regression model —hereafter referred to as the random intercept multilevel binomial logistic model. Two levels were utilised—the individual level (representing level 1) and the community level (representing level 2). The level-1 units were the individual women, and the level-2 units were the communities, with the former nested within the latter. The equations for this model type can be found elsewhere [43,44]. The multilevel analysis takes into account the possible contextual nature of health outcomes and the complex sampling structure of the 2018 NDHS, thereby producing more accurate regression estimates and significance testing [45–47].

## Statistical analysis

All statistical analyses, including the preliminary stages, were performed using Stata version 15.0.

**Preliminary data preparation stage.** First, level weights, which corresponded to level-1 weights for the level-1 units and level-2 weights for the level-2 units, were approximated following Elkasabi et al. [47]. This was done to adjust for the oversampling and undersampling procedures carried out by the NDHS team at the different sampling stages, together with adjusting for the different nonresponse rates at the different sampling stages [47]. The application of level weights will thus ensure the national representativeness of the analytical sample size and the results from this study [47]. Following this, the dataset for this study was declared to have a multistage survey design structure using Stata [47]. This allowed Stata to incorporate the 2018 NDHS sampling design—which includes the strata, the two levels of analysis used in this study, and the estimated level-1 and level-2 weights—in the statistical analysis [47]. This was done using the 'svyset' command in Stata [47]. The commands for the multilevel models were then written with the 'svy' prefix—which is the 'svy: melogit' command [47].

After this, a detailed independent variable selection process was carried out following the adaptation of the recommendation by Bursac et al. [48]. This reduces the chances of having regression estimates that are numerically unstable and reduces the risk of having a type II error—this error is a result of the increased risk of having large standard errors (SERs) and hence a wide confidence interval (CI) range [48]. For the first step in the independent variable selection process, a multilevel univariable binomial logistic regression analysis was performed by regressing each of the individual-level independent variables against the dependent variable, with statistical significance set at $p < 0.25$ [48]. Therefore, any individual-level independent variable with all of its subcategories having a p value$\geq 0.25$ was dropped at this point from further analysis [48].

For the second step, a preliminary multilevel multivariable regression analysis was performed using all the individual-level independent variables remaining after the first step, with statistical significance set at $p < 0.10$. Therefore, any individual-level independent variable with all of its subcategories having a p value$\geq 0.10$ was dropped one after the other, with the preliminary multilevel multivariable regression analysis run again after each variable was dropped. This approach helped in detecting a possible confounding effect—which was defined as a change of more than 20% in any of the regression estimates of the individual-level independent variables left in the model after each variable with a p value$\geq 0.10$ was removed from the regression analysis one after the other [48]. Therefore, if a change of more than 20% in the regression estimate was observed after dropping a particular individual-level independent variable during this step, this variable was retained for further analysis [48].

For the third step, those individual-level independent variables that were previously excluded after the first step were then included in another preliminary multilevel regression one after the other, together with all the individual-level independent variables remaining after

the second step. This procedure was used to determine whether any of the individual-level independent variables that were previously excluded after the first step would have a p value<0.10 [48]. If this occurred, then it was retained for further statistical analysis; otherwise, it was excluded [48]. Note that the first, second, and third variable selection steps were repeated again but now for the selection of only the community-level independent variables.

For the fourth step, as part of the independent variable selection process, a correlation test was carried out for all the individual and community-level variables remaining after the third variable selection step. For the correlation test, a coefficient cut-off point of ≥0.6, as seen from other empirical papers such as Midi et al. [49] and Yaya et al. [50], was used to detect the pairs of independent variables that were highly correlated. One of the variables from each highly correlated pair was then dropped. This independent variable selection step helps to reduce the occurrence of multicollinearity [49,50]. The final selected individual and community-level variables were subsequently obtained after the correlation test.

**Descriptive statistics.**   Descriptive statistical analysis, using the number of observations and percentages, was carried out for all the final selected independent variables, both at the individual level and the community level, across both subcategories of the dependent variable.

**Multilevel model-building process.**   Note that for all of the multilevel statistical analyses, the statistical significance level was set at a p value<0.05, with the regression effects of the independent variables, which are the fixed effects, presented as odds ratios (ORs)/adjusted odds ratios (aORs) with accompanying SERs, 95% CIs, and p values [43]. Following Sommet and Morselli [43] in relation to the random intercept multilevel model-building steps, first, the null or the empty model (Model A) was built, followed by the model with only the final selected individual-level variables adjusted for (Model B). Model C, which adjusted for only the final selected community-level variables, was subsequently built. Finally, Model D (the final model) was built, which adjusted for both the final selected individual- and community-level variables.

Summary measures such as the intracluster correlation coefficient (ICC)/variance partition coefficient (VPC), proportion of change in variance (PCV), and median odds ratio (MOR) were used to further explain the general contextual effect—that is, the measure of clustering or variation [43,45,46]. Further summary measures, such as the proportion of opposed odds ratios (POOR) and interval odds ratios (IORs), were used to explain in greater detail the fixed regression effects of the community-level variables [45,51,52]. Finally, two different postestimation diagnostic tests were carried out: sensitivity analysis and the goodness of fit test using the Wald test [47,53]. The sensitivity analysis helped to assess the reliability of the regression results, and this was done in this study by switching to DHS examples of modern methods. These are the modern methods listed above but now also including the LAM and SDM [5]. All the methods outside of these were thus classified as non-modern methods for the purpose of the sensitivity analysis.

## Results

### Descriptive statistics

Note that the results for the first, second, and third steps of the independent variable selection process are not shown. However, in relation to the correlation test, the variable representing women's highest formal education was highly correlated with the following variables: the highest formal education attained by husbands/partners (correlation coefficient = 0.65); the household wealth index (correlation coefficient = 0.60); and the community level of education for women (correlation coefficient = 0.62), and vice versa. The variable 'household wealth index' was also highly correlated with the variable 'community level of education for women'

(correlation coefficient = 0.56) and vice versa. A decision was thus made to remove the following variables from further analysis: women's highest formal education and the household wealth index. This allowed for the variables representing female and male educational levels to be included in this study, either at the individual or community level. The final selected independent variables are shown in Table 1. Table 1 also shows the results of the descriptive statistical analysis.

From the results in Table 1, 31.0% of the married or in-union WRA had mDFPS, leaving 69.0% of them with non-mDFPS. With respect to the age of the women as an individual-level variable, the highest and lowest percentages of the married or in-union WRA with DFP were between 35 and 39 years (21.7%) and 15 and 19 years (2.8%), respectively. Generally, a greater percentage of married or in-union WRA within the relatively older age groups had mDFPS than did those within the relatively younger age groups. For example, of the 2.8% of married or in-union WRA aged 15 to 19 years with a DFP, only 12.5% had mDFPS, leaving 87.5% of them with non-mDFPS. On the other hand, of the 8.6% of the married or in-union WRA aged 45 to 49 with a DFP, 32.6% had mDFPS, leaving 67.4% of them with non-mDFPS. Therefore, there is a possible positive association between increasing age of the women and increased mDFPS. Similar interpretations follow for the other independent variables in Table 1. Further inferential statistical analysis is thus needed to establish whether all of these possible associations are real effects or if they occurred by chance.

## Multilevel models

Table 2 shows the results for the multilevel models.

The Wald test results in Table 2 show that the inclusion of the additional variables in each of the successive models in comparison to the models without these additional variables significantly improved the fit of each of the successive models. Model D was, thus, the final chosen model. In addition, when comparing Model A to Model D, it was observed that the inclusion of all the final selected individual and community-level independent variables in Model D explained approximately 42.8% of the total variance estimated in Model A in relation to the odds of the WRA having their mDFPS, as opposed to non-mDFPS.

From the constant (intercept) obtained from Model D, it was estimated that across all the communities, thereby giving the overall intercept effect for a typical married or in-union WRA belonging to a typical community, there is an approximate reduction of 99.5% in the overall odds/likelihood of these women having their mDFPS, as opposed to non-mDFPS. This effect is thus based on the random intercept variance (RIV) being equal to zero. Furthermore, since Model D included all the final selected individual and community-level variables, this intercept effect was thus for married or in-union WRA with characteristics represented as each of their reference subcategories at the same time—for example, married or in-union women aged 15 to 19 years who were also from the Hausa/Fulani/Kanuri ethnic group, and so on. There is 95% confidence that in the population, across all the communities, there was an approximate reduction by between 98.4% and 99.9% in the odds of these particular WRA having their mDFPS, as opposed to non-mDFPS (aOR = 0.005; 95% CI = 0.001 to 0.016; p<0.001). This intercept effect was therefore, statistically significant.

However, from the estimate of the RIV, it was observed that this intercept effect significantly differed between communities, thereby giving each community a community-specific intercept (RIV = 0.719; 95% CI = 0.571 to 0.905). The large value of the RIV thus indicates that the married or in-union WRA have a greater likelihood of having their mDFPS, as opposed to non-mDFPS, in some communities than in others. The MOR and ICC further explain the level of variation and clustering, respectively, in terms of the odds of the outcome variable

**Table 1. Descriptive statistics results.**

|  | mDFPS, n (%) | non-mDFPS, n (%) | Total, N (%) |
|---|---|---|---|
|  | 2,830(31.0) | 6,292 (69.0) | 9, 122 (100) |
| *Individual-level (level 1) independent variables* | | | |
| **Age group of women:** | | | |
| 15 to 19 | 32 (12.5) | 223 (87.5) | 255 (2.8) |
| 20 to 24 | 271 (25.7) | 783 (74.3) | 1,054 (11.6) |
| 25 to 29 | 543 (30.9) | 1,214 (69.1) | 1,757 (19.3) |
| 30 to 34 | 593 (30.6) | 1,342(69.4) | 1,935 (21.2) |
| 35 to 39 | 667 (33.7) | 1,314(66.3) | 1,981 (21.7) |
| 40 to 44 | 467 (34.5) | 885 (65.5) | 1,352 (14.8) |
| 45 to 49 | 257 (32.6) | 531(67.4) | 788 (8.6) |
| **Ethnicity:** | | | |
| Hausa/Fulani/Kanuri | 480(20.8) | 1,830 (79.2) | 2,310 (25.3) |
| Yoruba | 772(42.3) | 1,055 (57.7) | 1,827 (20.0) |
| Igbo | 414(26.0) | 1,177 (74.0) | 1,591 (17.4) |
| Others | 1,164(34.3) | 2,230 (65.7) | 3,394 (37.2) |
| **Total number of children alive:** | | | |
| 0 child alive | 19 (15.3) | 105 (84.7) | 124 (1.4) |
| 1 to 4 children alive | 1,783 (32.3) | 3,729 (67.7) | 5,512 (60.4) |
| more than 4 children alive | 1,028 (29.5) | 2,458(70.5) | 3,486 (38.2) |
| **Ideal number of children:** | | | |
| 0 child | 36 (19.0) | 153 (81.0) | 189 (2.1) |
| 1 to 4 children | 1,137 (36.9) | 1,945 (63.1) | 3,082 (33.8) |
| more than 4 children | 1,657 (28.3) | 4,194 (71.7) | 5,851 (64.1) |
| **Highest level of formal education attained by husbands/partners:** | | | |
| no formal education | 284 (14.4) | 1,691 (85.6) | 1,975 (21.7) |
| primary education | 429 (29.8) | 1,011 (70.2) | 1,440 (15.8) |
| secondary education | 1, 342(35.1) | 2,481(64.9) | 3,721 (40.8) |
| higher education | 775(41.1) | 1,109(58.9) | 1,884 (20.7) |
| **Employment status of the women in the last 12 months prior to the survey, type of earnings, and the control of the earnings:** | | | |
| no, not employed and those employed, but either not paid at all or paid in kind | 775(25.9) | 2,216(74.1) | 2,991 (32.8) |

(*Continued*)

**Table 1.** (Continued)

| | mDFPS, n (%) | non-mDFPS, n (%) | Total, N (%) |
|---|---|---|---|
| | **2,830(31.0)** | **6,292 (69.0)** | **9, 122 (100)** |
| yes, employed and paid in cash, but the control of cash earnings mainly done by the husbands/partners | 132(27.6) | 347 (72.4) | 479 (5.3) |
| yes, employed and paid in cash, but the control of cash earnings done only by the women. | 1,349(32.9) | 2,756(67.1) | 4,105 (45.0) |
| yes, employed and paid in cash, but decisions about the cash earnings jointly done by the husbands/partners and the women | 574(37.1) | 973(62.9) | 1,547 (17.0) |
| **Age of women at first cohabitation or marriage:** | | | |
| <18 years | 1, 024(27.2) | 2,741(72.8) | 3,765 (41.3) |
| 18 to 30 years | 1, 734(34.0) | 3,370 (66.0) | 5,104 (56.0) |
| 31 to 49years | 72(28.5) | 181(71.5) | 253(2.8) |
| **Money for medical care:** | | | |
| yes, it is a big problem | 1,012(25.7) | 2,926(74.3) | 3,938 (43.2) |
| no, it is not a big problem | 1,818(35.1) | 3,366(64.9) | 5,184 (56.8) |
| **Exposure to family planning messages through media sources (radio, television, text messages to phones, and newspapers or magazines):** | | | |
| no exposure | 1, 463(27.2) | 3,923(72.8) | 5,386 (59.0) |
| exposed to only one media source | 609(32.1) | 1, 286 (67.9) | 2, 084 (20.8) |
| exposed to two to three media sources | 672(40.5) | 986(59.5) | 1,658 (18.2) |
| exposed to all the media sources | 86(47.0) | 97(53.0) | 183(2.0) |
| **Visitation by field workers, and a possible discussion about family planning, in the past 12 months before the survey:** | | | |
| no | 2,243(30.0) | 5,239(70.0) | 7,482 (82.0) |
| yes, but family planning was not discussed | 324(34.2) | 623(65.8) | 947(10.4) |
| yes, and family planning was discussed | 263(38.0) | 430(62.0) | 693(7.6) |
| **Decision making on visitation to families and relatives:** | | | |
| husband/partner alone | 685(22.7) | 2,327(77.3) | 3, 012 (33.0) |
| women alone | 490(33.9) | 957(66.1) | 1,447 (15.9) |
| joint decision making between women and their husbands/partners | 1,655(35.5) | 3,008(64.5) | 4,663 (51.1) |
| *Community-level (level 2) independent variables* | | | |
| **Place of residence:** | | | |
| rural | 1,300(26.7) | 3,570(73.3) | 4,870 (53.4) |
| urban | 2,722(64.0) | 1,530(36.0) | 4,252 (46.6) |
| **Community median age at first marriage:** | | | |
| low | 492 (20.9) | 1,867(79.1) | 2,359 (25.9) |

(*Continued*)

**Table 1.** (Continued)

| | mDFPS, n (%) | non-mDFPS, n (%) | Total, N (%) |
|---|---|---|---|
| | **2,830(31.0)** | **6,292 (69.0)** | **9, 122 (100)** |
| high | 2,338(34.6) | 4,425(65.4) | 6,763 (74.1) |
| **Community median ideal number of children:** | | | |
| high | 1,573(27.7) | 4,113(72.3) | 5,686 (62.3) |
| low | 1,257(36.6) | 2,179(63.4) | 3,436 (37.7) |
| **Community level of education for women:** | | | |
| low | 964(23.7) | 3,104(76.3) | 4,068 (44.6) |
| high | 1,866(36.9) | 3,188(63.1) | 5,054 (55.4) |
| **Community employment level for women:** | | | |
| low level employment | 525(24.1) | 1,656(75.9) | 2,181 (23.9) |
| high level employment | 2,305(33.2) | 4,636(66.8) | 6,941 (76.1) |
| **Community wealth level:** | | | |
| high poverty level | 857(24.2) | 2,682(75.8) | 3,539 (38.8) |
| low poverty level | 1,973(35.3) | 3,610(64.7) | 5,583 (61.2) |
| **Communities within states with CRLs, or not:** | | | |
| yes—presence of CRLs | 669(24.6) | 2,052(75.4) | 2,721 (29.8) |
| no—absence of CRLs | 2,161(33.8) | 4,240(66.2) | 6,401 (70.2) |
| **Community exposure of women to family planning messages through at least one of the different media sources (radio, television, text messages to phones, and newspapers or magazines):** | | | |
| low | 913(25.7) | 2,639(74.3) | 3,552 (38.9) |
| high | 1,917(34.4) | 3,653(65.6) | 5,570 (61.1) |
| **Geopolitical zones:** | | | |
| North Central zone | 709(37.9) | 1,162(62.1) | 1,871 (20.5) |
| North East zone | 321(21.6) | 1,164(78.4) | 1,485 (16.3) |
| North West zone | 404(29.3) | 973(70.7) | 1,377 (15.1) |
| South East zone | 303(24.0) | 960(76.0) | 1,263 (13.8) |
| South West zone | 755(39.1) | 1,177(60.9) | 1,932 (21.2) |

(*Continued*)

**Table 1.** (Continued)

|  | mDFPS, n (%) | non-mDFPS, n (%) | Total, N (%) |
|---|---|---|---|
|  | **2,830(31.0)** | **6,292 (69.0)** | **9, 122 (100)** |
| South South zone | 338(28.3) | 856(71.7) | 1,194 (13.1) |

Note: All of the estimations are unweighted.

non-mDFPS = demand for family planning satisfied with non-modern methods or not satisfied with any method at all, whether modern or non-modern; mDFPS = demand for family planning satisfied with modern methods; n = number; % = percentage; and CRLs = cultural and religious laws. The CRLs considered in this study were the presence of Sharia law and the non-domestication of the Child Rights Act (CRA). According to McGovern et al. [28] and Adebowale-Tambe [54], the 12 states in Nigeria with CRLs in operation are Bauchi, Yobe, Adamawa, Borno, and Gombe, all of which are in the North East geopolitical zone; Sokoto, Zamfara, Katsina, Kaduna, Kebbi, Jigawa, and Kano, all of which are in the North West geopolitical zone.

Source: Authors' computation using Stata version 15.0.

occurring. From the MOR estimate, it was found from Model D that if two married or in-union WRA with identical characteristics in terms of their individual-level characteristics were randomly sampled from two different communities, one community with married or in-union women having a relatively greater likelihood of having mDFPS, as opposed to non-mDFPS, in comparison to the other, the former woman is approximately 2.245 times more likely (in the median sense) to have her mDFPS, as opposed to non-mDFPS, in comparison to the latter woman.

The MOR is a measure of variation between different communities in relation to the odds of the married or in-union women having their mDFPS, as opposed to non-mDFPS, which is not explained by all the adjusted independent variables. From the results of this study, the MOR value is large indicating that there are other factors, both at the individual and community levels, that explain between-community variation in the odds of mDFPS, as opposed to non-mDFPS, that were not accounted for by all the adjusted independent variables in Model D. This finding also explains the large RIV and the PCV of less than 50% obtained in this model. On the other hand, from the ICC estimate in Model D, there is an approximately 17.9% within-community correlation or homogeneity in the odds of the married or in-union WRA having their mDFPS, as opposed to non-mDFPS. Furthermore, according to the VPC, approximately 17.9% of the total variation (both within-and between-community variation) in the odds of the married or in-union WRA having their mDFPS, as opposed to non-mDFPS, occurred between the communities. This means that approximately 82.1% of the total variation (both within- and between-community variation) in the odds of the married or in-union WRA having their mDFPS, as opposed to non-mDFPS, occurred within the communities—that is, between the individual women residing within a community.

From the fixed regression effect of the community-level variables obtained from Model D, it was observed that there was a statistically significant greater odds of having mDFPS, as opposed to non-mDFPS, among the married or in-union WRA in the communities with a high education level compared to those in communities with a low education level by a factor of 1.584, while holding all other fixed regression effects and the random effect constant (aOR = 1.584; 95% CI = 1.259 to 1.991; p<0.001). This is the overall average aOR for the fixed regression effect of the variable 'community level of education for women'. Note that in this

**Table 2. Results for the multilevel models.**

| Independent variables | ORs(95% CI) | SERs | p value | aORs (95%CI) | SERs | p value | aORs (95% CI) | SERs | p value | aORs (95%CI) | SERs | p value |
|---|---|---|---|---|---|---|---|---|---|---|---|---|
| | **Model A** | | | **Model B** | | | **Model C** | | | **Model D** | | |
| | | | | *Individual-level (level 1) variables* | | | | | | | | |
| **Age group of women:** | | | | | | | | | | | | |
| 15 to 19(ref.) | | | | 1.00 | | | | | | 1.00 | | |
| 20 to 24 | | | | 1.507(0.886 to 2.562) | 0.408 | 0.130 | | | | 1.479(0.869 to 2.517) | 0.401 | 0.149 |
| 25 to 29 | | | | 1.887(1.117 to 3.189) | 0.505 | 0.018** | | | | 1.818(1.074 to 3.078) | 0.488 | 0.026** |
| 30 to 34 | | | | 1.954(1.146 to 3.330) | 0.531 | 0.014** | | | | 1.870(1.098 to 3.186) | 0.508 | 0.021** |
| 35 to 39 | | | | 2.164(1.238 to 3.782) | 0.616 | 0.007*** | | | | 2.060(1.178 to 3.604) | 0.587 | 0.011** |
| 40 to 44 | | | | 2.270(1.304 to 3.951) | 0.641 | 0.004*** | | | | 2.133(1.224 to 3.716) | 0.603 | 0.008*** |
| 45 to 49 | | | | 1.940(1.018 to 3.698) | 0.638 | 0.044** | | | | 1.830(0.958 to 3.493) | 0.603 | 0.067* |
| **Ethnicity:** | | | | | | | | | | | | |
| Hausa/Fulani/Kanuri (ref.) | | | | 1.00 | | | | | | 1.00 | | |
| Yoruba | | | | 2.112(1.583 to 2.818) | 0.310 | <0.001*** | | | | 2.484(1.654 to 3.731) | 0.515 | <0.001*** |
| Igbo | | | | 0.945(0.666 to 1.339) | 0.168 | 0.749 | | | | 1.341(0.807 to 2.229) | 0.347 | 0.257 |
| others | | | | 1.490(1.136 to 1.953) | 0.206 | 0.004*** | | | | 1.798(1.277 to 2.532) | 0.314 | 0.001*** |
| **Total number of children alive:** | | | | | | | | | | | | |
| 0 children alive (ref.) | | | | 1.00 | | | 1.00 | | | | | |
| 1 to 4 children alive | | | | 2.140(1.001 to 4.574) | 0.828 | 0.050** | | | | 2.104(0.996 to 4.443) | 0.801 | 0.051* |
| more than 4 children alive | | | | 2.114(0.958 to 4.664) | 0.852 | 0.064* | | | | 2.124(0.976 to 4.622) | 0.842 | 0.057* |
| **Ideal number of children:** | | | | | | | | | | | | |
| 0 child (ref.) | | | | 1.00 | | | | | | 1.00 | | |
| 1 to 4 children | | | | 2.011(1.185 to 3.413) | 0.542 | 0.010** | | | | 1.989(1.177 to 3.362) | 0.532 | 0.010** |
| more than 4 children | | | | 1.681(1.009 to 2.799) | 0.437 | 0.046** | | | | 1.643(0.991 to 2.726) | 0.424 | 0.054* |
| **Highest level of formal education attained by the husbands/partners:** | | | | | | | | | | | | |
| no formal education (ref.) | | | | 1.00 | | | | | | 1.00 | | |
| primary education | | | | 2.001(1.513 to 2.646) | 0.285 | <0.001*** | | | | 1.949(1.474 to 2.577) | 0.277 | <0.001*** |
| secondary education | | | | 2.357(1.801 to 3.084) | 0.323 | <0.001*** | | | | 2.193(1.670 to 2.879) | 0.304 | <0.001*** |
| higher education | | | | 2.854(2.139 to 3.808) | 0.419 | <0.001*** | | | | 2.539(1.896 to 3.399) | 0.377 | <0.001*** |
| **Age of women at first cohabitation or marriage:** | | | | | | | | | | | | |
| <18 years (ref.) | | 1.00 | | | | | | 1.00 | | | | |
| 18 to 30 years | | | | 0.857(0.720 to 1.020) | 0.076 | 0.082* | | | | 0.840(0.705 to 1.000) | 0.075 | 0.050* |
| 31 to 49years | | | | 0.451(0.286 to 0.712) | 0.105 | 0.001*** | | | | 0.447(0.284 to 0.704) | 0.103 | 0.001*** |

*(Continued)*

**Table 2.** (Continued)

| Independent variables | ORs(95% CI) | SERs | p value | aORs (95%CI) | SERs | p value | aORs (95% CI) | SERs | p value | aORs (95%CI) | SERs | p value |
|---|---|---|---|---|---|---|---|---|---|---|---|---|
| **Money for medical care:** | | | | | | | | | | | | |
| yes, it is a big problem (ref.) | | | | 1.00 | | | | | | 1.00 | | |
| no, it is not a big problem | | | | 1.250(1.063 to 1.469) | 0.103 | 0.007 *** | | | | 1.212(1.031 to 1.424) | 0.100 | 0.020** |
| **Exposure to family planning messages through different media sources (radio, television, text messages to phones, and newspapers or magazines):** | | | | | | | | | | | | |
| no exposure (ref.) | | 1.00 | | | | | | 1.00 | | | | |
| exposed to only one media source | | | | 1.117(0.942 to 1.326) | 0.097 | 0.204 | | | | 1.067(0.897 to 1.268) | 0.094 | 0.466 |
| exposed to two to three media sources | | | | 1.261(1.044 to 1.525) | 0.122 | 0.016 ** | | | | 1.217(1.004 to 1.475) | 0.119 | 0.045 ** |
| exposed to all the media sources | | | | 1.877(1.198 to 2.942) | 0.430 | 0.006 *** | | | | 1.851(1.185 to 2.892) | 0.421 | 0.007 *** |
| **Decision making on visitation to families and relatives:** | | | | | | | | | | | | |
| husband/partner alone (ref.) | | | | 1.00 | | | | | | 1.00 | | |
| women alone | | | | 1.350(1.065 to 1.712) | 0.163 | 00.013** | | | | 1.300(1.025 to 1.648) | 0.157 | 0.030** |
| joint decision making between women and husband/partner | | | | 1.271(1.073 to 1.506) | 0.110 | 0.006 *** | | | | 1.263(1.065 to 1.498) | 0.110 | 0.007 *** |
| **Visitation by field workers, and a possible discussion about family planning, in the past 12 months before the survey:** | | | | | | | | | | | | |
| no(ref.) | | | | 1.00 | | | | | | 1.00 | | |
| yes, but family planning was not discussed | | | | 1.239(0.999 to 1.535) | 0.135 | 0.051* | | | | 1.223(0.986 to 1.516) | 0.134 | 0.067* |
| yes, and family planning was discussed | | | | 1.214(0.942 to 1.566) | 0.157 | 0.135 | | | | 0.447(0.938 to 1.557) | 0.156 | 0.144 |
| **Employment status of the women in the last 12 months prior to the survey, type of earnings, and the control of the earnings:** | | | | | | | | | | | | |
| no, not employed and those employed, but either not paid at all or paid in kind (ref.) | | | | 1.00 | | | | | | 1.00 | | |
| yes, employed and paid in cash, but the control of cash earnings mainly done by the husbands/partners | | | | 1.074(0.767 to 1.504) | 0.184 | 0.678 | | | | 1.097(0.783 to 1.537) | 0.18900.189 | 0.591 |
| yes, employed and paid in cash, but the control of cash earnings done only by the women. | | | | 1.079(0.893 to 1.303) | 0.104 | 0.430 | | | | 1.058(0.873 to 1.282) | 0.103 | 0.564 |
| yes, employed and paid in cash, but decision about the cash earnings jointly done by the women and their husbands/partners. | | | | 1.310(1.056 to 1.625) | 0.144 | 0.014 ** | | | | 1.314(1.058 to 1.632) | 0.145 | 0.013** |
| *Community-level (level 2) variables* | | | | | | | | | | | | |
| **Place of residence:** | | | | | | | | | | | | |
| rural (ref.) | | | | | | | 1.00 | | | 1.00 | | |
| Urban | | | | | | | 1.235 (1.039 to 1.469) | 0.109 | 0.017 ** | 1.180(0.992 to 1.403) | 0.104 | 0.062* |
| POOR (%);80% IOR | | | | | | | 43.3; 0.239 to 6.374 | | | 44.4; 0.254 to 5.485 | | |
| **Community median age at first marriage/cohabitation:** | | | | | | | | | | | | |
| low(ref.) | | | | | | | 1.00 | | | 1.00 | | |

*(Continued)*

**Table 2.** (Continued)

| Independent variables | ORs(95% CI) | SERs | p value | aORs (95%CI) | SERs | p value | aORs (95% CI) | SERs | p value | aORs (95%CI) | SERs | p value |
|---|---|---|---|---|---|---|---|---|---|---|---|---|
| high | | | | | | | 1.515 (1.165 to 1.971) | 0.203 | 0.002 *** | 1.220(0.934 to 1.595) | 0.167 | 0.145 |
| POOR (%);80% IOR | | | | | | | 37.5; 0.294 to 7.817 | | | 43.3; 0.263 to 5.674 | | |
| **Community median ideal number of children:** | | | | | | | | | | | | |
| high (ref.) | | | | | | | 1.00 | | | 1.00 | | |
| low | | | | | | | 0.844 (0.695 to 1.026) | 0.084 | 0.088* | 0.779(0.638 to 0.951) | 0.079 | 0.014** |
| POOR (%);80% IOR | | | | | | | 44.8; 0.164 to 4.355 | | | 41.7; 0.168 to 3.623 | | |
| **Community level of education for women:** | | | | | | | | | | | | |
| low(ref.) | | | | | | | 1.00 | | | 1.00 | | |
| high | | | | | | | 2.015 (1.597 to 2.541) | 0.238 | <0.001 *** | 1.584(1.259 to 1.991) | 0.185 | <0.001 *** |
| POOR (%);80% IOR | | | | | | | 29.1; 0.390 to 10.394 | | | 35.2; 0.341 to 7.362 | | |
| **Communities within states with CRLs, or not:** | | | | | | | | | | | | |
| yes(ref.) | | | | | | | 1.00 | | | 1.00 | | |
| no | | | | | | | 2.811 (1.341 to 5.890) | 1.060 | 0.006 *** | 1.853(0.907 to 3.788) | 0.675 | 0.091* |
| POOR (%);80% IOR | | | | | | | 20.9; 0.545 to 14.502 | | | 30.5; 0.399 to 8.616 | | |
| **Community exposure of women to family planning messages through at least one of the different media sources (radio, television, text messages to phones, and newspapers or magazines):** | | | | | | | | | | | | |
| low (ref.) | | | | | | | 1.00 | | | 1.00 | | |
| high | | | | | | | 1.371 (1.133 to 1.660) | 0.133 | 0.001 *** | 1.230(1.012 to 1.494) | 0.122 | 0.038** |
| POOR (%);80% IOR | | | | | | | 40.1; 0.266 to 7.076 | | | 43.3; 0.265 to 6.056 | | |
| **Community wealth index:** | | | | | | | | | | | | |
| high poverty level (ref.) | | | | | | | 1.00 | | | 1.00 | | |
| low poverty level | | | | | | | 1.239 (1.013 to 1.515) | 0.127 | 0.037** | 1.103(0.901 to 1.350) | 0.114 | 0.343 |
| POOR (%);80% IOR | | | | | | | 43.2; 0.240 to 6.390 | | | 46.8; 0.237 to 5.126 | | |
| **Community level of employment for women:** | | | | | | | | | | | | |
| low employment level (ref.) | | | | | | | 1.00 | | | 1.00 | | |

(*Continued*)

**Table 2.** (*Continued*)

| Independent variables | ORs(95% CI) | SERs | p value | aORs (95%CI) | SERs | p value | aORs (95% CI) | SERs | p value | aORs (95%CI) | SERs | p value |
|---|---|---|---|---|---|---|---|---|---|---|---|---|
| high employment level | | | | | | | 1.566 (1.259 to 1.948) | 0.174 | <0.001 *** | 1.209(0.964 to 1.517) | 0.140 | 0.100 |
| POOR (%);80% IOR | | | | | | | 36.3; 0.304 to 8.081 | | | 43.6; 0.260 to 5.622 | | |
| **Geopolitical zones:** | | | | | | | | | | | | |
| North Central zone(ref.) | | | | | | | 1.00 | | | 1.00 | | |
| North East zone | | | | | | | 1.418 (0.693 to 2.903) | 0.518 | 0.339 | 1.366(0.683 to 2.730) | 0.482 | 0.377 |
| POOR (%); 80% IOR | | | | | | | 39.4; 0.275 to 7.317 | | | 39.7; 0.294 to 6.237 | | |
| North West zone | | | | | | | 2.084 (0.952 to 4.563) | 0.832 | 0.066* | 2.477(1.152 to 5.330) | 0.967 | 0.020** |
| POOR (%);80% IOR | | | | | | | 28.4; 0.404 to 10.752 | | | 22.4; 0.533 to 11.517 | | |
| South East zone | | | | | | | 0.243 (0.189 to 0.314) | 0.032 | <0.001 *** | 0.337(0.214 to 0.532) | 0.078 | <0.001 *** |
| POOR (%); 80%IOR | | | | | | | 13.6; 0.047 to 1.256 | | | 18.1; 0.073 to 1.568 | | |
| South West zone | | | | | | | 0.659 (0.512 to 0.849) | 0.085 | 0.001 *** | 0.579(0.427 to 0.786) | 0.090 | <0.001 *** |
| POOR (%);80% IOR | | | | | | | 37.1; 0.128 to 3.401 | | | 32.3; 0.125 to 2.693 | | |
| South South zone | | | | | | | 0.420 (0.326 to 0.542) | 0.054 | <0.001 *** | 0.454(0.348 to 0.593) | 0.062 | <0.001 *** |
| POOR (%); 80% IOR | | | | | | | 24.8; 0.081 to 2.168 | | | 25.5; 0.098 to 2.112 | | |
| **Constant** | 0.346 (0.318 to 0.376) | 0.015 | <0.001 *** | 0.012 (0.004 to 0.033) | 0.006 | <0.001*** | 0.052 (0.024 to 0.115) | 0.021 | <0.001 *** | 0.005(0.001 to 0.016) | 0.003 | <0.001 *** |
| *Random effect* | | | | | | | | | | | | |
| | Estimate (95%CI) | SERs | p value | Estimate (95%CI) | SERs | p value | Estimate (95%CI) | SERs | p value | Estimate (95%CI) | SERs | p value |
| **Random intercept variance (RIV)** | 1.257 (1.050 to 1.505) | 0.115 | NA | 0.855 (0.677 to 1.081) | 0.102 | NA | 0.820 (0.671 to 1.001) | 0.084 | NA | 0.719(0.571 to 0.905) | 0.085 | NA |
| *General contextual effect* | | | | | | | | | | | | |
| **ICC (%) = VPC (%)** | 27.6 | | | 20.6 | | | 20.0 | | | 17.9 | | |
| **PCV (%)** | reference model | | | 32.0 | | | 34.8 | | | 42.8 | | |
| **MOR** | 2.914 | | | 2.416 | | | 2.372 | | | 2.245 | | |
| *Postestimation diagnosis* | | | | | | | | | | | | |

(*Continued*)

**Table 2.** (Continued)

| Independent variables | ORs(95% CI) | SERs | p value | aORs (95%CI) | SERs | p value | aORs (95% CI) | SERs | p value | aORs (95%CI) | SERs | p value |
|---|---|---|---|---|---|---|---|---|---|---|---|---|
| **Goodness of fit (Wald test)** | | | | | | | | | | | | |
| F statistics | reference model | | | model B vs. model A = 11.69 | | | model C vs. model A = 24.48 | | | model D vs. model A = 10.88; model D vs. model B = 7.32; and model D vs. model C = 7.67 | | |
| p value | reference model | | | <0.001*** | | | <0.001*** | | | <0.001***; <0.001***;<0.001*** | | |

Note: Ref. = the reference subcategory for each variable; ORs = odds ratios; aOR = adjusted odds ratios; 95% CI = 95% confidence interval; SERs = standard errors; NA = not available; ICC = intracluster correlation coefficient; VPC = variance partition coefficient; PCV = proportion of change in variance; MOR = median odds ratio; POOR = proportion of opposed odds ratios; 80% IOR = 80% interval odds ratios; vs. = versus/in comparison with; CRLs = cultural and religious laws.

'*' = significant at the 10% level (p < 0.10)

'**' = significant at the 5% level (p < 0.05); and

'***' = significant at the 1% level (p < 0.01).

Source: Authors' computation using Stata version 15.0.

study, a community with a high education level for women indicates that the percentage of the married or in-union women therein with at least a secondary school education is at least greater than the estimate at the national level. The national-level estimate is the percentage of WRA in Nigeria, regardless of marital status, with at least secondary school education, as estimated using data from all the WRA, regardless of their marital status, sampled by the 2018 NDHS.

However, the results for the 80% IOR and POOR showed that this regression effect was heterogeneous. In relation to this, it was observed from the POOR results that in approximately 35.2% of the comparisons between communities with a high education level and communities with a low education level for women, the aOR will be in the opposite direction to the overall aOR for this specific variable. That is, in 35.2% of these pairwise comparisons, there would be reduced odds of the married or in-union women having their mDFPS, as opposed to non-mDFPS, among those in communities with high educational levels compared to those in communities with low educational levels. Furthermore, still owing to this regression result's heterogeneity, according to the results of the 80% IOR, upon comparing different pairs of married or in-union women, the odds of having mDFPS, as opposed to non-mDFPS, will lie between 0.341 and 7.362 in the middle 80% of the distribution of the aORs when these comparisons are done. These pairs of women being compared have identical characteristics in terms of the other individual and community-level independent variables, but with one selected from a community with married or in-union women having a high educational level and the other from a community with married or in-union women having a low educational level.

Note that the wide 80% IOR, which included the null odds ratio value '1', is not surprising given the large RIV obtained in this study. The wide 80% IOR thus reflects considerable uncertainty in the regression effect of the community education level for women on mDFPS due to the substantial residual variation in the odds of mDFPS, as opposed to non-mDFPS, between communities that was not accounted for by all the independent variables adjusted for in Model D. The residual cluster variability is thus large in comparison to the regression effect of this particular community-level variable. Similar interpretations follow for the fixed regression effects of the other community-level variables.

Furthermore, at the individual level, it was found that upon holding all the other fixed regression effects and the random effect constant, the relatively older married or in-union women had statistically significant greater odds of having their mDFPS, as opposed to non-mDFPS, in comparison to those 15 to 19 years old, except for those 45 to 49 and 20 to 24 years old. For example, the married or in-union women who were 40 to 44 years old were approximately 2.133 times more likely to have their mDFPS, as opposed to non-mDFPS, in comparison to those 15 to 19 years old (aOR = 2.133; 95% CI = 1.224 to 3.716; p = 0.008). There is 95% confidence that in the population, this regression effect ranged from an increased odds of approximately 1.224 to 3.716, thereby implying a statistically significant regression effect. Similar interpretations follow for the fixed regression effects of the other individual-level variables.

On comparing the MOR with the fixed regression effects of the individual-level variables since they are all measured on the OR scale, it was found that the residual variance between communities that was not accounted for by all the independent variables adjusted for in Model D, as measured by MOR = 2.245, was of greater relevance than the fixed regression effects of most of the individual-level independent variables. This is all in relation to increasing the odds of the married or in-union WRA having their mDFPS, as opposed to non-mDFPS. This is because the MOR value of 2.245 was greater than the aORs for the fixed regression effect of most of the individual-level variables, excluding that for women's partners with higher (that is, more than secondary school) formal education (MOR = 2.245 vs. aOR = 2.539, respectively) and that for women with Yoruba ethnicity (MOR = 2.245 vs. aOR = 2.484, respectively).

Table 3 shows the results for model Ds, which was the final model for the sensitivity analysis. It was observed that there is no statistically significant difference between the RIV obtained from Model D in Table 2 and Model Ds in Table 3 since their 95% CIs overlapped. The MOR and ICC were thus similar in both models. The same observations were made regarding the estimates obtained from most of the independent variables, including in relation to the heterogeneity of the regression effects of the community-level variables. However, again, it was found from Model Ds that the residual variance between communities that was not accounted for by all the independent variables adjusted for in the sensitivity analysis, as measured by MOR = 2.274, was of greater relevance than the impact of most of the individual-level independent variables. This is because the MOR value of 2.274 was greater than the aORs for the fixed regression effect of most of the individual-level variables, excluding those for women's partners with higher formal education (MOR = 2.274 vs. aOR = 2.326, respectively); women with one to four children alive (MOR = 2.274 vs. aOR = 2.614, respectively); and women with more than four children alive (MOR = 2.274 vs. aOR = 2.661, respectively). It was concluded that the results obtained from Model D in Table 2 were mostly not sensitive to the changes implemented and were thus reliable.

## Discussion

### Summary of findings

This study was an empirical analysis of the mDFPS concentrating on the married or in-union WRA in Nigeria, with data obtained from the 2018 NDHS, using a random intercept multilevel binomial logistic model. It was also conceptually based on the HBM. Only married or in-union women were included because several variables, such as household decision-making power in terms of economic decisions, social decisions, and decisions pertaining to reproductive health care, have been implicated in possibly increasing the MCU [26]. This is because these variables measure the power dynamics between women and their husbands/partners [26]. This is particularly important in Nigeria because of her patriarchal setting, with an

**Table 3. Results for model Ds[a].**

| Independent variables | aORs (95%CI) | SERs | p value |
|---|---|---|---|
| *Individual-level (level 1) variables* | | | |
| **Age group of women:** | | | |
| 15 to 19(ref.) | 1.00 | | |
| 20 to 24 | 1.359(0.843 to 2.189) | 0.330 | 0.207 |
| 25 to 29 | 1.768(1.110 to 2.817) | 0.420 | 0.016** |
| 30 to 34 | 1.624(1.005 to 2.624) | 0.397 | 0.047** |
| 35 to 39 | 1.700(1.002 to 2.887) | 0.459 | 0.049** |
| 40 to 44 | 1.677(1.009 to 2.788) | 0.434 | 0.046** |
| 45 to 49 | 1.375 (0.756 to 2.499) | 0.419 | 0.296 |
| **Ethnicity:** | | | |
| Hausa/Fulani/Kanuri (ref.) | 1.00 | | |
| Yoruba | 1.839(1.220 to 2.771) | 0.384 | 0.004*** |
| Igbo | 1.553(0.969 to 2.490) | 0.373 | 0.067* |
| others | 1.621(1.181 to 2.224) | 0.261 | 0.003*** |
| **Total number of children alive:** | | | |
| 0 children alive (ref.) | 1.00 | | |
| 1 to 4 children alive | 2.614(1.266 to 5.398) | 0.966 | 0.009*** |
| more than 4 children alive | 2.661(1.233 to 5.742) | 1.043 | 0.013** |
| **Ideal number of children:** | | | |
| 0 child (ref.) | 1.00 | | |
| 1 to 4 children | 2.163(1.302 to 3.594) | 0.560 | 0.003*** |
| more than 4 children | 1.873(1.137 to 3.085) | 0.476 | 0.014** |
| **Highest educational level attained by the husbands/partners:** | | | |
| no formal education (ref.) | 1.00 | | |
| primary education | 1.828(1.389 to 2.405) | 0.256 | <0.001*** |
| secondary education | 2.048(1.575 to 2.662) | 0.274 | <0.001*** |
| higher education | 2.326(1.659 to 3.260) | 0.400 | <0.001*** |
| **Age of women at first marriage/cohabitation:** | | | |
| <18 years (ref.) | 1.00 | | |
| 18 to 30 years | 0.944(0.797 to 1.118) | 0.081 | 0.505 |
| 31 to 49years | 0.476(0.304 to 0.745) | 0.109 | 0.001*** |
| **Money for medical care:** | | | |
| yes, it is a big problem (ref.) | 1.00 | | |

(*Continued*)

**Table 3.** (Continued)

| Independent variables | aORs (95%CI) | SERs | p value |
|---|---|---|---|
| *Individual-level (level 1) variables* | | | |
| no, it is not a big problem | 1.128(0.966 to 1.317) | 0.089 | 0.128 |
| **Exposure to family planning messages through various media sources (radio, television, text messages to phones, and newspapers or magazines):** | | | |
| no exposure(ref.) | 1.00 | | |
| exposed to only one media source | 1.023(0.860 to 1.217) | 0.090 | 0.797 |
| exposed to two to three media sources | 1.158(0.953 to 1.406) | 0.115 | 0.140 |
| exposed to all the media sources | 1.714(1.090 to 2.695) | 0.395 | 0.020** |
| **Decision making on visitation to family and relatives:** | | | |
| husband/partner alone(ref.) | 1.00 | | |
| women alone | 1.162(0.942 to 1.434) | 0.124 | 0.161 |
| joint decision making between women and husband/partner | 1.103(0.927 to 1.313) | 0.098 | 0.269 |
| **Visitation by field workers, and a possible discussion about family planning, in the past 12 months before the survey:** | | | |
| no(ref.) | 1.00 | | |
| yes, but family planning was not discussed | 1.239(0.995 to 1.543) | 0.139 | 0.055* |
| yes, and family planning was discussed | 1.291(1.008 to 1.653) | 0.163 | 0.043** |
| **Employment status of the women in the last 12 months prior to the survey, type of earnings, and the control of cash earnings:** | | | |
| no, not employed and those employed, but either not paid at all or paid in kind (ref.) | 1.00 | | |
| yes, employed and paid in cash, but the control of cash earnings mainly done by the husbands/partners | 1.054(0.773 to 1.438) | 0.167 | 0.739 |
| yes, employed and paid in cash, but the control of cash earnings done only by the women. | 1.039(0.885 to 1.219) | 0.085 | 0.641 |
| yes, employed and paid in cash, but decision about the cash earnings jointly done by the women and their husbands/partners. | 1.208(0.986 to 1.480) | 0.125 | 0.068* |
| *Community-level (level 2) variables* | | | |
| **Place of residence:** | | | |
| rural (ref.) | 1.00 | | |
| urban | 1.174(0.994 to 1.386) | 0.099 | 0.059* |
| POOR(%); 80%IOR | 44.8; 0.246 to 5.594 | | |
| **Community median age at first marriage/cohabitation:** | | | |
| low median age (ref.) | 1.00 | | |
| high median age | 1.236(0.954 to 1.600) | 0.163 | 0.108 |
| POOR(%); 80%IOR | 43.3; 0.259 to 5.889 | | |
| **Community median ideal number of children:** | | | |
| high median number (ref.) | | | |

(*Continued*)

**Table 3.** (*Continued*)

| Independent variables | aORs (95%CI) | SERs | p value |
|---|---|---|---|
| *Individual-level (level 1) variables* | | | |
| low median number | 0.871 (0.719 to 1.055) | 0.085 | 0.158 |
| POOR(%); 80%IOR | 45.6; 0.183 to 4.150 | | |
| **Community level of education for women:** | | | |
| low educational level (ref.) | 1.00 | | |
| high educational level | 1.608 (1.289 to 2.006) | 0.181 | <0.001*** |
| POOR(%); 80%IOR | 34.8; 0.337 to 7.662 | | |
| **Communities within states with CRLs, or not:** | | | |
| yes(ref.) | 1.00 | | |
| no | 1.258 (0.622 to 2.546) | 0.452 | 0.522 |
| POOR(%); 80%IOR | 42.5; 0.264 to 5.994 | | |
| **Community exposure of women to family planning messages through at least one of the different media sources (radio, television, text messages to phones, and newspapers or magazines):** | | | |
| low exposure level (ref.) | 1.00 | | |
| high exposure level | 1.255 (1.031 to 1.527) | 0.126 | 0.024** |
| POOR(%); 80%IOR | 42.5; 0.263 to 5.980 | | |
| **Community wealth index:** | | | |
| high poverty level(ref.) | 1.00 | | |
| low poverty level | 1.067 (0.876 to 1.300) | 0.107 | 0.521 |
| POOR(%); 80%IOR | 48.0; 0.224 to 5.084 | | |
| **Community level of employment for women:** | | | |
| low employment level(ref.) | 1.00 | | |
| high employment level | 1.111(0.891 to 1.385) | 0.125 | 0.349 |
| POOR(%); 80%IOR | 46.4; 0.233 to 5.294 | | |
| **Geopolitical zones:** | | | |
| North Central zone(ref.) | 1.00 | | |
| North East zone | 1.373 (0.685 to 2.752) | 0.486 | 0.371 |
| POOR(%); 80%IOR | 39.7; 0.288 to 6.542 | | |
| North West zone | 1.552 (0.733 to 3.287) | 0.594 | 0.251 |
| POOR(%); 80%IOR | 35.9; 0.326 to 7.395 | | |
| South East zone | 0.360 (0.236 to 0.550) | 0.078 | <0.001*** |
| POOR(%); 80%IOR | 27.8; 0.076 to 1.715 | | |

(*Continued*)

**Table 3.** (Continued)

| Independent variables | aORs (95%CI) | SERs | p value |
|---|---|---|---|
| *Individual-level (level 1) variables* | | | |
| South West zone | 0.756 (0.558 to 1.023) | 0.117 | 0.070* |
| POOR(%); 80%IOR | 40.9; 0.159 to 3.602 | | |
| South South zone | 0.482 (0.371 to 0.625) | 0.064 | <0.001*** |
| POOR(%); 80%IOR | 27.4; 0.101 to 2.297 | | |
| **Constant** | 0.008 (0.002 to 0.026) | 0.005 | <0.001*** |
| *Random effect* | | | |
| | **Estimate (95% CI)** | SERs | |
| **Random intercept variance (RIV)** [RIV for model As[b] = 1.162(0.973 to 1.387), SERs = 0.105] | 0.742 (0.600 to 0.918) | 0.080 | NA |
| *General contextual effect* | | | |
| **ICC (%) = VPC (%)** | 18.4 | | |
| **PCV(%)** | 36.1 | | |
| **MOR** | 2.274 | | |
| *Postestimation diagnosis* | | | |
| **Goodness of fit (Wald test)** | | | |
| F statistics | Model Ds vs. model As = 10.29 | | |
| p value | <0.001*** | | |

Note: Ref. = the reference subcategory for each variable; aOR = adjusted odds ratios; 95% CI = 95% confidence interval; SERs = standard errors; NA = not available; ICC = intracluster correlation coefficient; VPC = variance partition coefficient; PCV = proportion of change in variance; MOR = median odds ratio; POOR = proportion of opposed odds ratios; 80% IOR = 80% interval odds ratios; vs. = versus/in comparison; and CRLs = cultural and religious laws.

'*' = significant at 10% level (p < 0.10)

'**' = significant at 5% level (p < 0.05); and

'***' = significant at 1% level (p < 0.01).

[a]Final model for the sensitivity analysis, which was carried out using the Demographic and Health Survey list of modern contraceptive methods; and [b]null model for the sensitivity analysis, which was also carried out using the Demographic and Health Survey list of modern contraceptive methods.

Source: Authors' computation using Stata version 15.0.

associated possibly reduced significance in terms of women's active participation in both the formal and informal settings where decisions are made and where resources are owned [55].

From our empirical literature search, this is the first inferential statistical analysis of mDFPS carried out among Nigerian married or in-union WRA. From this study, it was found that there is a between-community variance in the odds of these women having their mDFPS, as opposed to non-mDFPS, as seen from the RIV, ICC, and MOR estimates obtained from the null model. However, after adjusting for both the individual and community-level variables ultimately selected, approximately 42.8% of this between-community variance was accounted for. This means that identifying and then properly addressing these variables through policies will reduce the between- and within-community variance in the odds of women having their

mDFPS, as opposed to non-mDFPS. This will thus culminate in an increase in the national mDFPS estimate.

It was also found from the results of this study at the individual level that the relatively older married or in-union women, especially those 25 to 44 years, generally had statistically significant greater odds of having their mDFPS, as opposed to non-mDFPS, in comparison to those within the youngest age group (15 to 19 years). This finding is consistent with that obtained by Gichangi et al. [29] in Kenya and by Some et al. [31] in Burkina Faso. This may be because according to the 2018 NDHS report, older women, in comparison to relatively younger women, generally have a relatively higher education level, relatively higher empowerment level in terms of decision-making agency, and are more likely to have paid employment and greater access to and control of resources [5]. All of these variables have been significantly associated with an increased likelihood of women having their mDFPS, as opposed to having non-mDFPS [29,31].

This is because these variables, as seen from the adaptation of the HBM by Hall [33], function as enabling factors to help women positively modify both their perceptions of the threat of an unintended pregnancy occurring and the cost-benefits when MC are used, with an associated increased likelihood of adopting MC among women with an NFP. Furthermore, these variables address some of the demand-side barriers implicated in the suboptimal use of MC in Nigeria [56–59]. These include women's reduced autonomy in terms of reproductive health care; limited financial capacity; and limited accessibility to family planning services due to, for instance, the cost of family planning services; women's limited autonomy in terms of freedom of movement; and distance from health care centres [56–59].

Therefore, not surprisingly, from the results of this study at the individual level, it was found that the following married or in-union women had statistically significant greater odds of having their mDFPS in opposition to having non-mDFPS: those for whom the money for accessing health care was not a problem compared to those for whom it was a problem; women who could decide either alone or jointly with their spouses about visitation to friends and relatives compared to those not involved at all in this decision-making; and women earning in cash together with having joint decision-making power in terms of its spending compared to those not working/those working but who were either not paid at all or paid in kind.

Furthermore, from the results of this study at the individual level, the younger and oldest married or in-union women aged 20 to 24 and 45 to 49 years, respectively, each had greater odds of having their mDFPS, as opposed to non-mDFPS, in comparison to those aged 15 to 19 years, although neither of these regression effects reached statistical significance. This finding suggested that the likelihoods of having mDFPS among married or in-union WRA aged 20 to 24 years and 45–49 years were each not significantly different from that among those aged 15 to 19 years. This finding is consistent with that obtained by Gichangi et al. [29] in Kenya. According to OlaOlorun and Hindin [59], older women wrongly believe that they are not at risk of becoming pregnant due to their perceived waning fertility or outright infertility and therefore do not need to use contraceptives even when they are still sexually active. However, since there is an increased risk of pregnancy-related mortality and morbidity in mothers and/or children among the older women compared to younger women, there is an immense need for protection from unwanted pregnancies through the use of MC when there is an identified NFP among older women [4].

Additionally, from the results of this study at the individual level, married or in-union women exposed to FP messages through two to three media sources or through all the different media sources each had statistically significant greater odds of having their mDFPS, as opposed to non-mDFPS, with the odds increasing as the number of media sources exposed to increased. This is all in comparison to those with no exposure at all to FP messages through

any of the media sources. The media sources considered in this study were radio, television, text messages to phones, and newspapers or magazines. Again, these findings are consistent with those obtained by Gichangi et al. [29] in Kenya and by Some et al. [31] in Burkina Faso.

In addition, from the results of this study at the community level, married or in-union women in communities with high exposure status to FP messages from at least one media source had statistically significant greater odds of having their mDFPS, as opposed to non-mDFPS, in comparison to those in communities with low exposure status. Note that in this study, a community with high exposure status to FP messages connotes that the percentage of the married or in-union women therein with exposure to FP messages from at least one media source is at least greater than the estimate at the national level. The national-level estimate is the percentage of WRA in Nigeria, regardless of marital status, with exposure to FP messages from at least one media source as estimated using data from all the WRA, regardless of their marital status, sampled by the 2018 NDHS. However, there was heterogeneity in this regression result. According to the 2018 NDHS report, women in Nigeria with low media exposure to FP messages are mostly in rural areas, compared to those in urban areas, and in the NE and NW geopolitical zones, compared to those in the NC, SW, SE, and SS zones [5].

Exposure to FP messages, at both the individual and community levels, will help married or in-union women become knowledgeable about who actually has an NFP, the importance of using MC to meet this need, and the various health centres or hospitals within their localities where these modern methods can be obtained [60]. Using the HBM, as adapted by Hall [33], exposure to FP messages through various media sources, at both the individual and community levels, will serve as an external stimulus cueing these women toward the action of positively modifying their perceptions of both the threat of unintended pregnancy occurring and the cost-benefits of using MC. This is because women become knowledgeable through exposure to FP messages about the diverse health and economic benefits of FP. This will thus all result in an increased likelihood of married or in-union women with an NFP adopting MC.

Married or in-union women exposed to FP messages, at both the individual and community levels, will thus help to address some of the demand-side barriers implicated in the suboptimal use of MC in Nigeria [56–59]. These include the following: misconceptions/myths about FP and fertility rate, which could be personal, cultural, or religious, such as the use of MC promoting infidelity/unfaithfulness; misconceptions about having an NFP, especially among women who are not engaging in sexual intercourse regularly; fear of side effects of contraceptive methods, especially modern methods; and lack of knowledge about the benefits of using MC and where to access FP services [56–59].

Additionally, from the results of this study at the individual level, married or in-union women with an ideal number of children of one to four had statistically significant greater odds of having mDFPS, as opposed to non-mDFPS, in comparison to those with an ideal number of children of zero. On the other hand, married or in-union women with an ideal number of children of more than four had greater odds of having their mDFPS, as opposed to non-mDFPS, in comparison to those with an ideal number of children of zero, although this regression effect did not reach statistical significance. This finding shows that a low fertility preference of less than four, as represented by the ideal number of children, is significantly associated with increased use of MC by the married or in-union WRA with a DFP. Additionally, it shows that the likelihood of having mDFPS among married or in-union women with an ideal number of children of more than four was not significantly different from that among the women with an ideal number of children of zero. Using the HBM, as adapted by Hall [33], women with low fertility preferences can serve as an enabling factor that positively modifies their perceptions of both the threat of unintended pregnancy occurring and the cost–benefits

of using MC. This will thus result in an increased likelihood of women with an NFP adopting MC.

However, in relation to the actual fertility rate, it was found from the results of this study at the individual level that married or in-union women with one to four children alive and those with more than four children alive each had greater odds of having their mDFPS, as opposed to non-mDFPS, in comparison to those with no children alive, although neither of these regression effects reached statistical significance. This, again, implies that the likelihoods of having mDFPS among married or in-union women with one to four children alive and those with more than four children alive were each not significantly different from that among women with no children alive. This finding contradicts the findings in rural Jordan by Komasawa et al. [30]. These findings in relation to the fertility rate may be due to Nigeria's strong pronatalist culture [61], which is reflected in her consistently high TFR of more than 5 live births per woman since 1990 [19].

On the other hand, unlike this study's findings about women's fertility preferences at the individual level, from the results at the community level, married or in-union women in communities with a low median ideal number of children had statistically significant lower odds of having their mDFPS, as opposed to non-mDFPS, in comparison to those in communities with a high ideal number of children. This regression effect was, however, heterogeneous. Note that in this study, a community with a low median ideal number of children connotes that the median ideal number of children for the married or in-union women therein is less than the national estimate. The national estimate is the median ideal number of children for all the WRA in Nigeria, regardless of marital status, using data from all the WRA, regardless of their marital status, sampled by the 2018 NDHS. Our findings thus show a possible discordance between the community norms of Nigerian married or in-union women having low fertility preferences and them actually taking an effective action through the use of MC to make this their low fertility preference a reality. This may be due to several demand and supply-side barriers that have been implicated in the suboptimal use of MC in Nigeria [56–59].

Notably, on the demand side, men in Nigeria consistently have a relatively higher fertility preference, as seen from several NDHS reports across the years [5,12–16], together with having relatively greater power/autonomy or a stronger influence on women's reproductive health, especially at the community level [5,12–16]. This is not surprising considering Nigeria's patriarchal structure [55]. Additionally, the contrast observed from this study's findings in relation to the regression effects of women's fertility preferences at the individual and community levels may show that within households, the covert use of MC may be very prevalent. This is a long-standing strategy known to be used by women to meet their NFP, especially within contexts where systemic power and gender norms put men in the dominating position in terms of reproductive decision-making, as seen within Nigeria [62,63].

On the other hand, notably on the supply side are the provision of poor quality services, especially from the public health centres, such as poor counselling services, poor interactions between health care providers and patients, and insufficient availability of health workers [56–59]. Additionally, the stockout of modern contraceptive methods commonly occurs in public health facilities in Nigeria [56–59]. This is particularly important in Nigeria because, according to the 2018NDHS, a slightly greater percentage of modern contraceptive users received contraceptives from the public health sector (54.0%), which was mainly from government health centres (28.5%), than from the private sector (40.8%), which was mainly from private chemists/patent medicine shops (21.5%) [5]. This was particularly true for highly cost-effective modern contraceptive methods such as the IUD, implants, and injectables [5]. This study's findings, in relation to the regression effect of the community median ideal number of children, however, require further investigation.

In relation to the effect of formal education, from the results of this study on the individual level, the married or in-union women with husbands/partners who were formally educated had statistically significant greater odds of having their mDFPS, as opposed to non-mDFPS, in comparison to those women with husbands/partners having no formal education, with the odds increasing as the level of formal education improved to the higher level (more than secondary education). This statistically significant positive regression effect is particularly important for women whose husbands/partners have higher education. This is because the results of our study showed that the positive regression effect of women's partners with higher education is more relevant in terms of increasing the likelihood of WRA having their mDFPS, as opposed to non-mDFPS, than the large unexplained between-community variance that was still present in the final model.

On the other hand, from the results of this study at the community level, it was found that the married or in-union women in communities with a high education level had a statistically significant greater odds of having their mDFPS, as opposed to non-mDFPS, in comparison to those in communities with women with a low education level. This association effect was, however, heterogeneous. This positive regression effect at the community level is consistent with that obtained by Some et al. [31] in Burkina Faso. According to the 2018 NDHS report, women in Nigeria with a low level of formal education are mostly in rural areas in comparison to those in urban areas and in the NE and NW geopolitical zones in comparison to those in the NC, SE, SW, and SS zones [5].

Our findings thus show that both married or in-union men and women formally educated in Nigeria, together with an associated increase in the level of formal education, will help increase the likelihood of women having their mDFPS. This is possibly because with education at the individual and community levels, both men and women will have greater health literacy skills due to increased access to various sources, such as media sources and health workers, together with an increased ability to receive and understand information from these sources [64,65]. This will thus help in addressing some demand-side barriers that have been implicated in Nigeria's suboptimal use of MC [56–59]. These include misconceptions/myth about FP on the side of both men and women; husbands'/partners' disapproval; lack of or limited or false knowledge about the use of MC; lack of or limited or false knowledge about where to access FP services; fear of side effects upon using MC due to experiences heard from other women in the communities; social norms favouring a high fertility rate, such as a strong son-preference culture; women's wrong personal perceived infecund status; and misconceptions about who actually has an NFP [56–59].

However, surprisingly, from the results of this study at the community level, it was observed that the married or in-union women in the NW geopolitical zone, in comparison to those in the NC zone, had statistically significant greater odds of having mDFPS, as opposed to non-mDFPS, although there was heterogeneity in this result. Furthermore, compared with women in the NC zone, married or in-union women in the NE geopolitical zone had greater odds of having their mDFPS, as opposed to non-mDFPS, but this regression effect did not reach statistical significance. This finding implies that the likelihood of having mDFPS among married or in-union women in the NE geopolitical zone is not significantly different from that among married or in-union women in the NC zone. There was also heterogeneity in this result. These positive regression effects may be due to the conflicts experienced in the Northern geopolitical zones, which increased in frequency in the NC and NE zones in 2018 (the year the 2018 NDHS was carried out), resulting in the loss of human life and disruption of human activities, possibly including access to health care [66]. However, further investigations are needed regarding these findings.

Furthermore, surprisingly, from the results of this study at the community level, it was observed that married or in-union women in the SS, SE, and SW geopolitical zones, in comparison to those in the NC, each had statistically significant lower odds of having their mDFPS, as opposed to non-mDFPS. However, there was heterogeneity in all of these results. These regression effects in these Southern geopolitical zones are consistent with those obtained by Ejembi et al. [26] in Nigeria using the contraceptive indicator 'mCPR' as the outcome variable. According to Ejembi et al. [26], this may be due to Southern geopolitical zones, especially the SE zone, having the largest population of Catholics.

The Catholic Church is known to be in opposition to the use of MC but instead supports the use of non-modern FP methods, especially fertility awareness-based methods [26]. This possibly explains our findings at the individual level concerning the positive regression effect of married or in-union women of the Igbo ethnicity, in comparison to those from the Hausa/Fulani/Kanuri ethnicity, on the mDFPS, as opposed to non-mDFPS, which did not reach statistical significance. This finding implies that the likelihood of having mDFPS among Igbo married or in-union women is not significantly different from that among Hausa/Fulani/Kanuri married or in-union women. Note that the Igbos are the main ethnic group in the SE geopolitical zone of Nigeria [37]. However, further investigations are still needed regarding this finding.

On the other hand, from the results of this study at the individual level, the married or in-union women from the Yoruba ethnicity and those from the other non-major ethnicity, each in comparison to those from the Hausa/Fulani/Kanuri ethnicity, each had a statistically significant positive regression effect on mDFPS, as opposed to non-mDFPS. However, from the regression effect of women's ethnicity, only the positive regression effect of Yoruba ethnicity was more relevant in terms of increasing the likelihood of women having their mDFPS, as opposed to non-mDFPS, than the large unexplained between-community variance that was present in the final model.

Furthermore, surprisingly, from the results of this study at the community level, married or in-union women residing in urban areas, in comparison to those in rural areas, had greater odds of having mDFPS, as opposed to non-mDFPS, but this regression effect did not reach statistical significance. This finding implies that the likelihood of having mDFPS among married or in-union women residing in urban areas is not significantly different from that among those residing in rural areas. There was also heterogeneity in this result. This regression effect is opposite to that found by Gichangi et al. [29] in Kenya. According to Okigbo et al. [67], although women residing in urban areas are more likely to utilise FP services than women in rural areas due to their greater accessibility, disparities do occur across the different wealth statuses among women in urban areas. Poor women residing within urban areas, especially those in urban slums, are said to have relatively worse reproductive health outcomes, including MCU, in comparison to their wealthier counterparts [67]. Nigeria is also said to have rapid urban population growth, which is expected to triple by 2050, more due to the high fertility rate within the urban areas, especially among the urban poor, than due to rural–urban migration [67]. All of these factors thus possibly explain our empirical findings in this study in relation to the effect of place of residence on mDFPS. However, further investigations are still needed regarding this finding.

## Limitations of the study

Despite this study filling a major gap in terms of the empirical analysis of mDFPS among Nigerian married or in-union women, our study has some limitations. First, this study used cross-sectional data, which prevents causal relationships from being established. Further research

using longitudinal data should thus be done. Furthermore, due to the use of self-reported data, there is an increased risk of response bias, as the respondents may provide answers that may be socially desirable rather than that representing the true scenario. However, this bias will be minimised through the DHS carefully selecting and training the interviewers used and through gender matching of the interviewers and respondents [68].

Additionally, this study is limited to Nigeria, and the outcome thus cannot be generalised globally. However, because the DHS has standardised methodology and data collection processes across countries and has consistent content over time, which in all also leads to the presence of similar variables in the DHS data across and within countries over time, this study can be easily replicated in other African or developing countries where DHS data are available [69]. This approach allows for comparisons of the results across countries and hence across different populations, both cross-sectionally and over time, and even within a country over time [69].

Future studies should also consider adjusting for other socioeconomic factors not included in this study's empirical model to possibly further account for the between-community variation in relation to the MCU among women with a DFP. These include factors relating to the physical accessibility of health centres/hospitals and the service provision quality within these facilities, both at the individual and community levels. Nevertheless, these findings from this study still provide major insights for policy makers, other researchers, and the public in relation to the topic of discussion.

## Policy implications

Community-based interventions should be organised by the Nigerian government, also involving community leaders and chiefs, to equally educate both men and women on the positive impact of women's equal participation in household decision making, including that pertaining to reproductive health care, freedom of movement, the use of personal income earned, and household income, on reproductive health. All of these factors will improve women's empowerment. These community-based interventions should also address harmful social and cultural reproductive practices, such as high fertility preferences (especially among men), high fertility rates, and age at first marriage/cohabitation of less than 18 years. These interventions should also aim at properly educating women and men on who actually has an NFP and the negative implications, both health and economic, of not having a met need, together with the immense importance of using MC to meet this need. All of the various media sources, both at the individual and community levels, should be extensively utilised by the Nigerian government to disseminate this information. The MC should also be made available within the nearest health centres to the communities at little or no cost.

Importantly, specific policy interventions should be implemented to improve the educational levels of both males and females, especially beyond the secondary school level, since it was found from this study that the large unexplained/residual between-community variation was not as important as the impact of women's partners having higher education in terms of increasing the odds of women satisfying their DFP with MC. This will also increase the employment potential, and thus the earning potential, of girls and women. For all of these community-based interventions, a greater concentration should be given to the following contexts: rural areas, but not excluding the urban areas due to heterogeneity found in the regression results; and the Northern geopolitical zones, but still not excluding the Southern zones due to the heterogeneity in the regression results.

Furthermore, within these specific contexts, the following married or in-union WRA should receive greater attention from the community-based interventions: young women aged

15 to 24 years; older women aged 45 to 49 years; women with Hausa/Fulani/Kanuri ethnicity; women with Igbo ethnicity; women without any formal education; women with husbands/partners without any formal education; women who are unable to access healthcare financially (including those who are not involved at all in the decision making regarding the use of their cash earnings); women with cultural limitations in terms of their freedom of movement and social interactions; women without any media access; women who are unemployed; women with high fertility preferences; and women with high fertility rates. Taken together, all of these will help to address the between- and within-community inequalities in terms of women satisfying their DFP with MC in Nigeria given her pluralistic nature, thereby culminating in an increase in the value of mDFPS nationally while looking towards the possible achievement of the SDG indicator 3.7.1 benchmark value by year 2030.

## Contributions to knowledge

From our empirical review, no other study has analysed the individual- and community-level factors associated with the mDFPS among Nigerian married or in-union women using inferential statistical analysis. This study thus attempted to fill this major gap and, in so doing, made important contributions to the body of knowledge, especially looking towards Nigeria's attainment of the SDG 3.7.1 benchmark by year 2030. It was found at the individual level that husbands'/partners' having higher education, in comparison to those with no formal education at all, significantly increased the odds of women having their mDFPS, and this regression effect was of greater relevance than the large residual between-community variance that was present in the final model as a result of unadjusted independent variables. This makes husbands'/partners' educational level a very important variable to address through government policies. This is an important finding, as the other empirical literature papers reviewed for this study have always concentrated more on females' education at the individual level.

Additionally, at the community level, it was found from the regression effect of the community median ideal number of children for women that there may be a possible discordance between the community social norms of women having low fertility preferences and the use of MC so as to make this low fertility preference a reality. This was, however, contrary to the finding at the individual level, where individual women with a low fertility preference of one to four children had significantly greater odds of using MC to satisfy their DFP. This could show a stronger community influence on reproductive decision-making by Nigerian men, who are known on average to consistently have a relatively greater fertility preference than women. All of these are features of a patriarchal society, such as Nigeria. This also shows that possibly within households, the covert use of MC may be very prevalent, which is also a characteristic of a patriarchal society. This makes reproductive healthcare decision-making power and community fertility preferences for both men and women important variables to address through the community-based interventions.

In addition, it was found that the regression effects of all the community-level variables were heterogeneous. This means that policies should be concentrated more within communities where, according to our findings, the women within have significantly lower odds of having their mDFPS, such as in communities with women with low educational levels. However, the counterpart communities, that is, those in which women within have a significantly greater odds of having their mDFPS, such as communities in which women have higher education levels, should not be overlooked. Other empirical research papers reviewed for this study, whether carried out inside or outside of Nigeria/Africa, did not explore the possible heterogeneity of the regression effects of the community-level variables in their multilevel analysis. Therefore, taken together, this study brought in some new dimensions in relation to the topic at hand.

## Conclusions

Nigeria's mDFPS among the married or in-union WRA is considered very low since it is less than 50%. Given the health and economic benefits of FP, an increase in the value of mDFPS is thus needed in Nigeria, while holding out hope towards her achieving the SDG indicator 3.7.1 benchmark of at least 75% in the mDFPS coverage by year 2030. From the findings of this study, this can possibly be achieved by improving the spread of information about the use of MC to meet the NFP through organised community interventions, together with the extensive use of all the various media outlets to spread this information. The empowerment of married or in-union women, especially in terms of their formal education level, employment status and cash earnings, and their active involvement in decision making in various aspects, including their personal income and reproductive health, should also be improved so as to increase the likelihood of them having their mDFPS. Women's husbands/partners with formal education, especially those with higher education levels, also increased the likelihood of women having their DFPS. Overall, the rural areas and the Northern geopolitical zones in Nigeria should receive more attention, but the urban areas and the Southern geopolitical zones should not be excluded.

## Acknowledgments

The authors are grateful to the DHS team for free and easy access to the NDHS dataset.

## Author Contributions

**Conceptualization:** Emomine Odjesa.

**Data curation:** Emomine Odjesa, Friday Ebhodaghe Okonofua.

**Formal analysis:** Emomine Odjesa, Friday Ebhodaghe Okonofua.

**Investigation:** Emomine Odjesa, Friday Ebhodaghe Okonofua.

**Methodology:** Emomine Odjesa, Friday Ebhodaghe Okonofua.

**Project administration:** Friday Ebhodaghe Okonofua.

**Resources:** Friday Ebhodaghe Okonofua.

**Software:** Emomine Odjesa.

**Supervision:** Friday Ebhodaghe Okonofua.

**Validation:** Friday Ebhodaghe Okonofua.

**Visualization:** Friday Ebhodaghe Okonofua.

**Writing – original draft:** Emomine Odjesa.

**Writing – review & editing:** Emomine Odjesa, Friday Ebhodaghe Okonofua.

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
