## [Decision Letter · Decision Letter 0]

6 Nov 2023

PONE-D-23-00022Manuscript Type: Original Research Article

An empirical analysis of the demand for family planning satisfied by modern methods among married or in-union women in Nigeria: Application of multilevel binomial logistic modelling technique.PLOS ONE

Dear Dr. Odjesa,

Thank you for submitting your manuscript to PLOS ONE. After careful consideration, we feel that it has merit but does not fully meet PLOS ONE’s publication criteria as it currently stands. Therefore, we invite you to submit a revised version of the manuscript that addresses the points raised during the review process.

We look forward to receiving your revised manuscript.

Kind regards,

Obasanjo Afolabi Bolarinwa, Masters

Academic Editor

PLOS ONE

Journal Requirements:

 Whilst you may use any professional scientific editing service of your choice, PLOS has partnered with both American Journal Experts (AJE) and Editage to provide discounted services to PLOS authors. Both organizations have experience helping authors meet PLOS guidelines and can provide language editing, translation, manuscript formatting, and figure formatting to ensure your manuscript meets our submission guidelines. To take advantage of our partnership with AJE, visit the AJE website (http://aje.com/go/plos) for a 15% discount off AJE services. To take advantage of our partnership with Editage, visit the Editage website (www.editage.com) and enter referral code PLOSEDIT for a 15% discount off Editage services. If the PLOS editorial team finds any language issues in text that either AJE or Editage has edited, the service provider will re-edit the text for free.

 A clean copy of the edited manuscript (uploaded as the new *manuscript* file)”".

5. Please ensure that you include a title page within your main document. You should list all authors and all affiliations as per our author instructions and clearly indicate the corresponding author.

Reviewers' comments:

Reviewer's Responses to Questions

**Comments to the Author**

1. Is the manuscript technically sound, and do the data support the conclusions?

Reviewer #1: Yes

Reviewer #2: Yes

2. Has the statistical analysis been performed appropriately and rigorously? 

Reviewer #1: Yes

Reviewer #2: Yes

3. Have the authors made all data underlying the findings in their manuscript fully available?

Reviewer #1: Yes

Reviewer #2: Yes

4. Is the manuscript presented in an intelligible fashion and written in standard English?

Reviewer #1: Yes

Reviewer #2: Yes

5. Review Comments to the Author

Reviewer #1: It is important to explain some more details when considering on some social context factors at community level that having statistically significant with dependent variable. Because they are very important for policy implementation.

Reviewer #2: The study focused on empirical analyses of demand for family planning satisfied by modern methods among married or in-union women in Nigeria. Please find below a few observations

1. What is MC and NFP (Line 142). Use the full before abbreviation to enable your readers follow through. The use of several abbreviation without earlier stating them in full is not helpful. Please look into it and correct

2. State the contribution of knowledge. This should come after the subheading :Policy Implications"

6. PLOS authors have the option to publish the peer review history of their article (what does this mean?). If published, this will include your full peer review and any attached files.

Reviewer #1: **Yes: **Yothin Sawangdee Ph.D (Sociology)

Reviewer #2: **Yes: **Chukwudeh Okechukwu Stephen

---

## [Author Response · Author response to Decision Letter 0]

4 Dec 2023

I am writing you this letter in order to respond to the comments made by reviewers and editor regarding my manuscript titled ‘An empirical analysis of the demand for family planning satisfied by modern methods among married or in-union women in Nigeria: Application of multilevel binomial logistic modelling technique’. According to the editors, major revisions were required. 

I have heeded to the comments made by the editors. In relation to this, I have made sure that the referencing, both in-text and referencing list, meets the PLOS ONE publishing standard. I have also made sure that the manuscript generally meets the recommended publishing guidelines, including that for structuring of figures and tables, of PLOS ONE. I have also done a thorough general English language editing using a language editing software called ‘curie’. The British English language was the language of choice used for the editing. I actually used this software twice to edit my manuscript.

I have also heeded to the comments made by the reviewers. Reviewer one asked for expansion of the discussion of the community level factors that were statistically significant, as this would be important for policy formation. In relation to this, I actually edited the entire sub-section labelled ‘summary of findings’. Reviewer two, on the other hand, asked for an initial explanation of all the abbreviations/acronyms used. This I have also carefully done. For the specific ones pointed out, ‘MC’ was first used and written in full in line 96, while NFP was first used and written in full in line 101. In addition, I have also included a sub-heading labelled ‘contributions to knowledge’ as part of the discussion section. This was also recommended by reviewer two. 

Also, I have changed the data availability statement on the application portal online to ‘ some restrictions apply to data availability’. This was done because accessing the data requires making a formal written request at the DHS website (https://dhsprogram.com/data/dataset/Nigeria_Standard-DHS_2018.cfm?flag=0). Upon getting a written approval, the data set can then be freely downloaded. An institutional/organisational access is thus needed.

---

## [Decision Letter · Decision Letter 1]

1 Feb 2024

PONE-D-23-00022R1Manuscript Type: Original Research Article

An empirical analysis of the demand for family planning satisfied by modern methods among married or in-union women in Nigeria: Application of multilevel binomial logistic modelling technique.PLOS ONE

Dear Dr. Odjesa,

Thank you for submitting your manuscript to PLOS ONE. After careful consideration, we feel that it has merit but does not fully meet PLOS ONE’s publication criteria as it currently stands. Therefore, we invite you to submit a revised version of the manuscript that addresses the points raised during the review process.

We look forward to receiving your revised manuscript.

Kind regards,

Jianhong Zhou

Staff Editor

PLOS ONE

Journal Requirements:

Reviewers' comments:

Reviewer's Responses to Questions

**Comments to the Author**

1. If the authors have adequately addressed your comments raised in a previous round of review and you feel that this manuscript is now acceptable for publication, you may indicate that here to bypass the “Comments to the Author” section, enter your conflict of interest statement in the “Confidential to Editor” section, and submit your "Accept" recommendation.

Reviewer #1: All comments have been addressed

Reviewer #2: All comments have been addressed

2. Is the manuscript technically sound, and do the data support the conclusions?

Reviewer #1: Yes

Reviewer #2: Yes

3. Has the statistical analysis been performed appropriately and rigorously? 

Reviewer #1: Yes

Reviewer #2: Yes

4. Have the authors made all data underlying the findings in their manuscript fully available?

Reviewer #1: Yes

Reviewer #2: Yes

5. Is the manuscript presented in an intelligible fashion and written in standard English?

Reviewer #1: Yes

Reviewer #2: Yes

6. Review Comments to the Author

Reviewer #1: This revision is acceptable. I agree with an explanation that have been providing. Therefore, I think we able to distribute this manuscript under the journal title.

Reviewer #2: The article address the low demand for family planning satisfied by modern method

Introduction: The rationale for the study was clearly explained as current mDFPS among married and in-union women is less than 50%. Please check your sentences to be sure you are not repeating words, sentence and ideas.

Method

A, what is MC, NFP etc. Always state the full meaning of word before abbreviating

B. Describe the setting and location of the study

C. state the eligibility criteria for your sample

D. Variables for the study was clearly outline

Discussion

Is the study among Nigeria women or among married and in-union women? Please be consistent as women could include single mothers etc. This confusion should be address

Limitation of the study

This study is limited to Nigeria and the outcome cannot be generalize globally. This should be stated as the limitation of the study

7. PLOS authors have the option to publish the peer review history of their article (what does this mean?). If published, this will include your full peer review and any attached files.

Reviewer #1: **Yes: **Yothin Sawangdee

Reviewer #2: **Yes: **Chukwudeh Okechukwu Stephen

---

## [Author Response · Author response to Decision Letter 1]

21 Feb 2024

Emomine Odjesa,

 Centre of Excellence in Reproductive Health Innovation (CERHI),

 University of Benin, Benin City, Edo State, Nigeria.

 19th February 2024

Jianhong Zhou,

Staff Editor,

PLOS ONE.

Dear Sir/ma,

Response to reviewers and editors

I am writing you this letter in order to respond to the comments made by reviewers and the staff editor regarding my manuscript titled ‘An empirical analysis of the demand for family planning satisfied by modern methods among married or in-union women in Nigeria: Application of multilevel binomial logistic modelling technique’. According to the editor, minor revisions are required. 

We have heeded to the comments made by the editors. In relation to this, I have made sure that the reference list is complete and correct. I have also made sure that the papers cited, and hence included in the reference list, are current; that is, no retracted papers were included. In relation to this, the reference number [21] was changed to a relevant current paper. I have also, again, done a thorough general English language editing using a language editing software called ‘curie’. The British English language was the language of choice used for the editing. 

We have also heeded to the comments made by the second reviewer. Reviewer two asked for an initial explanation of all the abbreviations/acronyms used. This I have also carefully done. For the specific ones pointed out, ‘MC’ was first used and written in full in line 96, while NFP was first used and written in full in line 101. In addition, I have also included a sub-heading labelled ‘study setting’ as part of the materials and method section to explain the setting and location of the study, as required by reviewer two to be done. Regarding stating the eligibility criteria of the sample, as required by the second reviewer to be done, several additional words were included under the already existing sub-heading ‘the study’s analytical sample’. From line 274 to 280, the following statement was included: Based on the aims of this study, concentration was done on the married or in-union WRA in Nigeria with a DFP (that is, the totality of NFP), who were also usual residents of the communities in which they were surveyed. This clearly depicts the eligibility criteria for the women to be included in this study and thus represents this study’s analytical sample. Note that only the usual residents of the communities surveyed by the 2018 NDHS were concentrated on in this study because it allowed for examining the possible contextual influences on the likelihood of Nigerian married or in-union WRA with a DFP using MC. 

Furthermore, from line 319 to 328, the following statement was included under the already existing sub-heading ‘the study’s analytical sample’: Additionally, since this study’s analytical sample was obtained from the 2018 NDHS and because of the sample design of this NDHS, as explained previously, this study’s analytical sample thus covers the entire territory of Nigeria. This analytical sample is therefore representative of the Nigerian population structure, together with its naturally occurring hierarchies. This thus allows for, using appropriate statistical analysis, the distinguishing of individual-level and community-level factors, together with the estimation of the different variances at the different levels, that can possibly influence the likelihood of married or in-union women having mDFPS. Also, it allows for, using appropriate statistical techniques, the generalisation of our result findings from this sample into the Nigerian population. Thus, this analytical sample can be used to meet the aims of this study.

In addition, the second reviewer mentioned that the limitation of the non-generalisation of the results, globally, should also be included, because the study was limited to only Nigeria. This was included in line 889 to 895 under the already existing sub-heading ‘limitation of the study’ as part of the discussion section: Additionally, this study is limited to Nigeria, and the outcome thus cannot be generalised globally. However, because the DHS has standardised methodology and data collection processes across countries and has consistent content over time, which in all also leads to the presence of similar variables in the DHS data across and within countries over time, this study can be easily replicated in other African or developing countries where DHS data are available[69]. This approach allows for comparisons of the results across countries and hence across different populations, both cross-sectionally and over time, and even within a country over time[69].

I have also carefully noted under the discussion, in fact in the entire length of this paper, that the study is based on married or in-union women, not just women, of reproductive ages in Nigeria, as mentioned to be done by the second reviewer. I also carefully checked the introduction section, in fact the entire length of this paper, to make sure I was not repeating words, sentences, or ideas, as mentioned to be done by the second reviewer. 

Also, as previously noted in the first rebuttal letter I sent, I changed the data availability statement on the application portal online to ‘some restrictions apply to data availability’. This was done because accessing the data requires making a formal written request at the DHS website (https://dhsprogram.com/data/dataset/Nigeria_Standard-DHS_2018.cfm?flag=0). Upon getting a written approval, the data set can then be freely downloaded. In fact, from the following link, you can clearly see the conditions required to use the DHS dataset: https://dhsprogram.com/data/terms-of-use.cfm

Thank you very much for taking time out to review my initial application, the first revision, and also this present revision.

Yours faithfully,

Emomine Odjesa.

---

## [Editor Report · Decision Letter 2]

5 Mar 2024

Manuscript Type: Original Research Article

An empirical analysis of the demand for family planning satisfied by modern methods among married or in-union women in Nigeria: Application of multilevel binomial logistic modelling technique.

PONE-D-23-00022R2

Dear Dr. Odjesa,

We’re pleased to inform you that your manuscript has been judged scientifically suitable for publication and will be formally accepted for publication once it meets all outstanding technical requirements.

Kind regards,

Yothin Sawangdee, Ph.D

Guest Editor

PLOS ONE
---

## [Editor Report · Acceptance letter]

11 Mar 2024

PONE-D-23-00022R2 

PLOS ONE

Dear Dr. Odjesa, 

I'm pleased to inform you that your manuscript has been deemed suitable for publication in PLOS ONE. Congratulations! Your manuscript is now being handed over to our production team.

Kind regards, 

on behalf of

Dr. Yothin Sawangdee 

Guest Editor

PLOS ONE